# GRADIENT DESCENT FOR MATRIX FACTORIZATION: UNDERSTANDING LARGE INITIALIZATION

## ABSTRACT

In deep learning practice, large random initialization is commonly used. Understanding the behavior of gradient descent (GD) with such initialization is both crucial and challenging. This paper focuses on a simplified matrix factorization problem, delving into the dynamics of GD when using large initialization. Leveraging a novel signal-to-noise ratio argument and an inductive argument, we offer a detailed trajectory analysis of GD from the initial point to the global minima. Our insights indicate that even with a large initialization, GD can exhibit incremental learning, which coincides with experimental observations.

## 1 INTRODUCTION

Understanding generalization and optimization in deep learning remains a pivotal and challenging area of research (Sun, 2019; Jakubovitz et al., 2019). Despite their vast model complexity, neural networks consistently exhibit remarkable generalization properties (Zhang et al., 2016). Conventional theories, such as uniform convergence, fall short in fully explaining this exceptional success, spurring a plethora of new research on generalization.

One influential line of research delves into the implicit bias of gradient-based methods (Vardi, 2023). It is believed in these works that gradient-based algorithms induce an implicit bias towards solutions that generalize well. Prominent examples include Soudry et al. (2018)'s work on logistic regression, Arora et al. (2019)'s work on deep matrix factorization, and Ji & Telgarsky (2018)'s work on deep linear networks, among many others.

This paper focuses on the implicit bias of gradient descent (GD) in matrix factorization. Matrix factorization acts as a simplified model for neural network study, mirroring the training of a two-layer linear network. Additionally, it is intrinsically linked to a range of engineering problems including matrix sensing, matrix completion, dictionary learning, and phase retrieval, among others (Chi et al., 2019). In recent years, researchers have studied various optimization facets of matrix factorization, encompassing topics like optimization landscape (Sun et al., 2016; 2018; Zhu et al., 2021), global convergence and the convergence rate of GD (Gunasekar et al., 2017; Ma et al., 2018; Chen et al., 2019), and the effects of random initialization (Stöger & Soltanolkotabi, 2021). General theories in non-convex optimization have also shed significant light on the matrix factorization problem. Notably, Lee et al. (2016) show that GD escapes saddle points almost surely under the strict saddle point condition. This implies the global convergence of GD for problems whose local minima are all global minima and whose saddle points are all strict.

Despite these advancements in matrix factorization, the theoretical understanding of GD with large initialization remains largely unexplored. Specifically, consider the symmetric matrix factorization problem

$$\boldsymbol{X}^* = \arg\min_{\boldsymbol{X} \in \mathbb{R}^{d \times r}} \|\boldsymbol{\Sigma} - \boldsymbol{X}\boldsymbol{X}^\top\|_{\mathrm{F}}^2, \tag{1}$$

where $\boldsymbol{\Sigma} \in \mathbb{R}^{d \times d}$ is a positive semi-definite matrix of rank at least $r$. The solutions of problem (1) are given by $\boldsymbol{X}^*\boldsymbol{X}^{*\top} = \boldsymbol{\Sigma}_r$, where $\boldsymbol{\Sigma}_r$ is the best rank $r$ approximation of $\boldsymbol{\Sigma}$. Finding such $\boldsymbol{X}^*$ poses a non-convex optimization challenge, and much research has been undertaken to understand GD's behavior in solving this problem or its variants. For instance, Zhu et al. (2021) demonstrate that problem (1) has no spurious local minima, possesses only strict saddle points, and satisfies a

local regularity condition. Such analysis implies that GD converges to the global minima almost surely, and the convergence is at a linear rate if initialized in a local region surrounding the global minima. The global minima are unknown in practice, so researchers also examine GD with inaccurate or random initialization. Stöger & Soltanolkotabi (2021) show that if using a sufficiently small initialization, then GD behaves like a spectral method in early iterations. Based on this, the authors establish the linear global convergence rate of GD with small initialization. Furthermore, using a similar argument, Jin et al. (2023) demonstrate an incremental learning phenomenon of GD with small initialization; Eigenvectors associated with larger eigenvalues are learned first. GD with small initialization has also been studied in other works such as Ma et al. (2022); Soltanolkotabi et al. (2023) and related references. Nevertheless, the behavior of GD when initialized with large values remains less understood.

Here we refer to $X_0 = \varpi N_0$ as large initialization if $\varpi$ is a positive constant independent of $d$ and $N_0$'s entries are independently distributed as $\mathcal{N}(0, \frac{1}{d})$. Correspondingly, small initialization refers to the case where $\varpi$ tends to zero as $d$ tends to infinity. Notably, when $d$ tends to infinity, the norm $\|X_0\|$ converges to a positive constant (or zero) for large (or small) initialization. Existing literature using small initialization typically assumes $\varpi$ be of order $d^{-\iota(\kappa)}$, where $\kappa > 1$ is the conditional number and $\iota(\cdot)$ is an increasing function with $\iota(\infty) = \infty$. Such small initialization, despite the solid theories, is seldom adopted in practice. For example, in deep learning, Lecun initialization (LeCun et al., 2002), Xavier initialization (Glorot & Bengio, 2010), Kaiming initialization (He et al., 2016), and many other initialization strategies all use large random initialization, i.e., $\varpi$ is a constant. Hence, despite the challenges, examining the properties of GD with large initialization is still of great importance.

This paper explores the behaviors of GD with large initialization when addressing problem (1). By using novel signal-to-noise (SNR) and inductive arguments, we offer a comprehensive analysis of the GD trajectory starting from the initial point to the global minima. We show that GD with large initialization may still exhibit an incremental learning phenomenon (Jin et al., 2023; Gissin et al., 2019; Li et al., 2020). Our result also implies the fast global convergence of GD under certain transition assumptions. It is worth noting that the verification of the transition assumptions remains a problem. For convenience, we informally summarize our results below.

**Theorem 1 (Informal)** *Suppose $\Sigma$ is a positive semi-definite matrix with leading $r + 1$ eigenvalues strictly decreasing. Let $X_t$ be the GD sequence for problem (1) with $X_0 = \varpi N_0$, where $\varpi$ is a positive constant independent of $d$ and $N_0 \in \mathbb{R}^{d \times r}$ has independent $\mathcal{N}(0, \frac{1}{d})$ entries. Then*

- *the GD sequence converges to the global minima almost surely (Lee et al., 2016; Zhu et al., 2021);*

- *a comprehensive trajectory analysis of GD is given, indicating that eigenvectors associated with larger eigenvalues are learned first;*

- *under an **unverified** transition assumption, GD achieves $\epsilon$-accuracy in $\mathcal{O}(\log(\frac{1}{\epsilon}) + \log(d))$ steps.*

To illustrate our results more clearly, we provide a simple but representative experiment on rank-two matrix approximation. The parameters are set as follows: $d = 4000$, $r = 2$, and $\Sigma = \text{diag}(1, 0.5, e)$, where $e \in \mathbb{R}^{d-r}$ is an arithmetic sequence transitioning from 0.3 down to 0. Let $X_0 = 0.5N_0$ with the entries of $N_0$ independently drawn from $\mathcal{N}(0, \frac{1}{d})$. We compute the GD sequence $X_t$ with a step size of 0.1 and evaluate the errors $\|\Sigma_r - X_t X_t^\top\|_F$, where $\Sigma_r = \text{diag}(1, 0.5, 0, \ldots, 0)$ is the best rank-$r$ matrix approximation to $\Sigma$. In Figure 1, we plot the error curve, highlight several noteworthy points on the curve, and depict the heat maps of the first three rows and columns of $X_t X_t^\top$ at these steps. Observations reveal that GD exhibits an incremental learning phenomenon and the error curve has two types of shapes: flat and steep.

To interpret the error curve displayed in Figure 1, we shall analyze the first $r$ rows of $X_t$ one by one. In particular, we will study the dynamics of the quantities $\sigma_1(u_{k,t})$ and $\sigma_1(u_{k,t} K_{k,t}^\top)$, where $u_{k,t}$ is the $k$-th row of $X_t$ and $K_{k,t}$ is the $(k+1)$-to-$d$-th rows of $X_t$. These quantities are associated with the diagonal and off-diagonal elements in the heat map of $X_t X_t^\top$. Hence, one can correspond our mathematical analysis with the dynamics of the heat maps displayed in Figure 1. Notably, our analysis on the SNR $\sigma_1^2(u_{k,t})/\sigma_1(u_{k,t} K_{k,t})$ demonstrates that the off-diagonal elements shall decrease in a geometric rate, once the signal strength $\sigma_1^2(u_{k,t})$ reaches a certain level. This motivates us to employ an inductive argument to analyze the whole convergence trajectory.

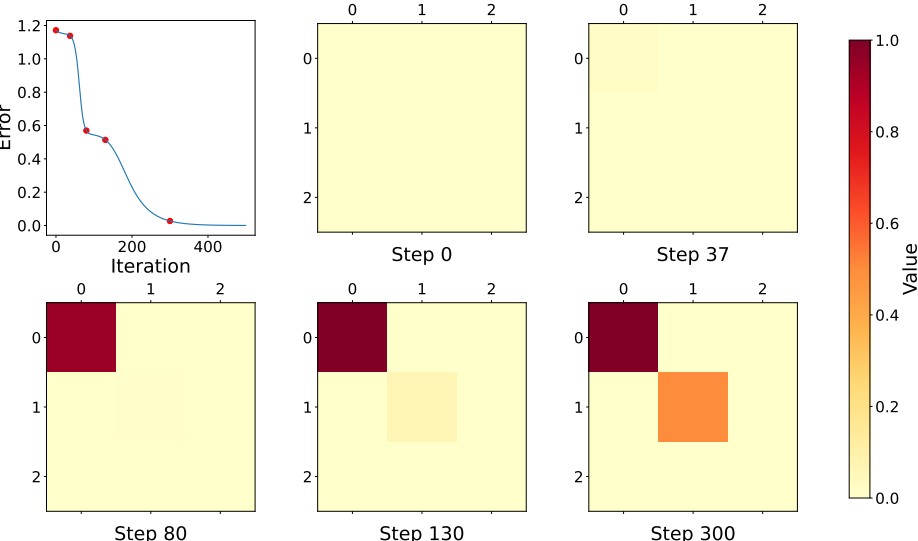

Figure 1: Plot 1 shows the errors $\|\boldsymbol{\Sigma}_r - \boldsymbol{X}_t\boldsymbol{X}_t^\top\|_{\mathrm{F}}$ over iterations. Plots 2-5 show the heat maps of the top three rows and columns of $\boldsymbol{X}_t\boldsymbol{X}_t^\top$ at iterations $t = 0, 37, 80, 140$, and $300$, corresponding to the red points in Plot 1.

The rest of this paper proceeds as follows. Section 2 reviews the usage of SNR analysis for rank-one matrix approximation. Section 3 uses the SNR analysis to prove the local linear convergence of GD in general rank problems. In Section 4, we examine the random initialization. Specifically, Section 4.1 reviews small initialization and Section 4.2 considers large initialization and presents our main theorem. In Section 5, we provide a sketch of proof. Concluding discussions are given in Section 6 and proofs are provided in the Appendix.

## 2 SNR ANALYSIS FOR RANK-ONE MATRIX APPROXIMATION

The rank-one matrix approximation is well-studied. Chen et al. (2019) demonstrated that GD with large random initialization exhibits linear convergence to the global minima, leveraging a SNR argument. Specifically, consider problem (1) with $r = 1$ and assume[1] $\boldsymbol{\Sigma} = \mathrm{diag}(\lambda_1, \ldots, \lambda_d)$ is diagonal with decreasing diagonal elements and $\lambda_1 > \lambda_2$. Let the initial point $\boldsymbol{x}_0 \in \mathbb{R}^d$ be a vector such that the first entry is non-zero and the norm $\|\boldsymbol{x}_0\|$ is smaller than $2\lambda_1$. Then $\boldsymbol{x}_t\boldsymbol{x}_t^\top$ converges to $\mathrm{diag}(1, 0, \ldots, 0)$ fast, where $\boldsymbol{x}_t$ is given by the GD update rule

$$\boldsymbol{x}_t = \boldsymbol{x}_{t-1} + \eta(\boldsymbol{\Sigma} - \boldsymbol{x}_{t-1}\boldsymbol{x}_{t-1}^\top)\boldsymbol{x}_{t-1}, \tag{2}$$

and $\eta$ is the learning rate. In their analysis, Chen et al. (2019) first decompose $\boldsymbol{x}_t$ as $\boldsymbol{x}_t = (a_t, \boldsymbol{b}_t)^\top$ with $a_t \in \mathbb{R}$ and $\boldsymbol{b}_t \in \mathbb{R}^{d-1}$. Then the GD rule can be rewritten as

$$a_t = a_{t-1} + \eta\lambda_1 a_{t-1} - \eta(a_{t-1}^2 + \|\boldsymbol{b}_{t-1}\|^2)a_{t-1}, \tag{3}$$

$$\boldsymbol{b}_t = \boldsymbol{b}_{t-1} + \eta\boldsymbol{\Sigma}_{\mathrm{res}}\boldsymbol{b}_{t-1} - \eta(a_{t-1}^2 + \|\boldsymbol{b}_{t-1}\|^2)\boldsymbol{b}_{t-1}, \tag{4}$$

where $\boldsymbol{\Sigma}_{\mathrm{res}} = \mathrm{diag}(\lambda_2, \ldots, \lambda_d)$. Let $\alpha_t = |a_t|$ and $\beta_t = \|\boldsymbol{b}_t\|$ and assume $\eta\lambda_1$ is smaller than some constant, say $\frac{1}{12}$. Then it is direct to derive that

$$\alpha_t = (1 + \eta\lambda_1 - \eta\alpha_{t-1}^2 - \eta\beta_{t-1}^2)\alpha_{t-1}, \tag{5}$$

$$\beta_t \leq (1 + \eta\lambda_2 - \eta\alpha_{t-1}^2 - \eta\beta_{t-1}^2)\beta_{t-1}. \tag{6}$$

Dividing (6) by (5), we can show that

$$\frac{\beta_t}{\alpha_t} \leq \frac{1 + \eta\lambda_2 - \eta\alpha_{t-1}^2 - \eta\beta_{t-1}^2}{1 + \eta\lambda_1 - \eta\alpha_{t-1}^2 - \eta\beta_{t-1}^2} \cdot \frac{\beta_{t-1}}{\alpha_{t-1}} \leq (1 - \frac{\eta\Delta}{3}) \cdot \frac{\beta_{t-1}}{\alpha_{t-1}}, \tag{7}$$

---

[1]There is no loss of generality to assume that $\boldsymbol{\Sigma}$ is diagonal because GD analysis is invariant to rotations.

where $\Delta = \lambda_1 - \lambda_2$ is the eigengap and the second inequality uses that

$$h(s) = \frac{1 - \eta\Delta/2 + s}{1 + \eta\Delta/2 + s} \le h(\frac{1}{2}) \le 1 - \frac{\eta\Delta}{3}, \quad \forall s \in [-\frac{1}{2}, \frac{1}{2}]. \tag{8}$$

Inequality (7) states that the ratio $\frac{\beta_t}{\alpha_t}$ will decay to zero geometrically fast. Using this, Chen et al. (2019) establish that $\beta_t$ and $\alpha_t$ converge fast to zero and $\lambda_1$ respectively. Our paper refers to this argument as a SNR analysis, and we refer to $\alpha_t$ as the signal strength and $\beta_t$ as the noise strength.

## 3 BENIGN INITIALIZATION

Generalizing the SNR argument to general rank problems poses additional challenges. For instance, the global minima cannot be characterized by the two real numbers $\alpha_t$ and $\beta_t$. Even if we find other effective quantities representing the GD sequence, giving desired dynamic analysis as in (5) and (6) remains challenging. In essence, this issue originates from the heterogeneity in different dimensions or mathematically the non-commutativity of matrix multiplication.

One way to tackle the issue is to use a benign initialization with a high initial SNR. This allows us to extend the SNR analysis to general rank problems and establish the local linear convergence of GD. Consider problem (1) with general $r$ and assume $\boldsymbol{\Sigma} = \mathrm{diag}(\lambda_1, \ldots, \lambda_d)$ is diagonal with decreasing diagonal elements and $\Delta := \lambda_r - \lambda_{r+1} > 0$. Let $\boldsymbol{X}_0 \in \mathbb{R}^{d \times r}$ be an initial point and

$$\boldsymbol{X}_t = \boldsymbol{X}_{t-1} + \eta(\boldsymbol{\Sigma} - \boldsymbol{X}_{t-1}\boldsymbol{X}_{t-1}^\top)\boldsymbol{X}_{t-1}, \tag{9}$$

where $\eta$ is the learning rate. For the SNR argument, we decompose $\boldsymbol{X}_t$ as $(\boldsymbol{U}_t^\top, \boldsymbol{J}_t^\top)^\top$, where $\boldsymbol{U}_t$ is the first $r$ rows of $\boldsymbol{X}_t$ and $\boldsymbol{J}_t$ is the rest $d - r$ rows of $\boldsymbol{X}_t$. In analogy to the rank-one case, we may think of $\boldsymbol{U}_t$ as the signal and $\boldsymbol{J}_t$ as the noise, because at the global minima $\boldsymbol{U}$ is non-zero while $\boldsymbol{J}$ is zero. By adopting a benign initialization, we mean $\sigma_r(\boldsymbol{U}_0)$ is large while $\sigma_1(\boldsymbol{J}_0)$ is small. More precisely, we define the following set

$$\mathcal{R} = \{\boldsymbol{X} = \begin{pmatrix} \boldsymbol{U} \\ \boldsymbol{J} \end{pmatrix} \mid \sigma_1^2(\boldsymbol{X}) \le 2\lambda_1, \sigma_r^2(\boldsymbol{U}) \ge \Delta/4, \sigma_1^2(\boldsymbol{J}) \le \lambda_r - \Delta/2\}. \tag{10}$$

The set $\mathcal{R}$ contains all the global minima of problem (1). Moreover, the SNR $\sigma_r^2(\boldsymbol{U})/\sigma_1^2(\boldsymbol{J})$ is larger than the constant $\Delta/(4\lambda_1)$ for any $\boldsymbol{X}$ in $\mathcal{R}$. If we initialize GD within $\mathcal{R}$, then the sequence $\boldsymbol{X}_t$ will remain in $\mathcal{R}$ and the SNR will grow fast to infinity. Consequently, we can establish the local linear convergence of GD as in Theorem 2. Theorem 2 is useful for examining random initialization. Specifically, when $\boldsymbol{X}_0 \notin \mathcal{R}$, the convergence of GD consists of two stages, the first stage when the sequence enters $\mathcal{R}$ and the final convergence stage. Only the first stage needs to be further analyzed.

**Theorem 2** *Suppose $\eta \le \frac{\Delta^2}{36\lambda_1^3}$, $\boldsymbol{X}_0 \in \mathcal{R}$, and $\boldsymbol{X}_t$ is given by (9). Then, for small $\epsilon > 0$, we have $\|\boldsymbol{\Sigma}_r - \boldsymbol{X}_t\boldsymbol{X}_t^\top\| \le \epsilon$ in $\mathcal{O}(\frac{6}{\eta\Delta} \ln \frac{200r\lambda_1^3}{\eta\Delta^2\epsilon})$ iterations, where $\boldsymbol{\Sigma}_r = \mathrm{diag}(\lambda_1, \ldots, \lambda_r, 0, \ldots, 0)$.*

**Remark 3** *While our paper aims to understand large initialization in later sections, Theorem 2 is still an additional contribution of the paper. Prior works on local linear convergence either study the rank-one case (Chen et al., 2019) or require $\boldsymbol{\Sigma}$ to be exact of rank $r$ (Zhu et al., 2021). Their arguments cannot be directly used to prove Theorem 2. In contrast, by employing an SNR argument, we can establish the local linear convergence for general cases. Our SNR analysis relies on a lower bound for the signal $\sigma_r^2(\boldsymbol{U}_{t+1})$ and an upper bound for the noise $\sigma_1^2(\boldsymbol{J}_{t+1})$. These two bounds need to be related so that the ratio of $SNR_{t+1}$ by $SNR_t$ can be analyzed. This is the challenging part of the SNR analysis. Finally, we note that although we assume $\boldsymbol{\Sigma}$ is positive semi-definite for simplicity, our proof can be easily extended to general symmetric $\boldsymbol{\Sigma}$. Also, it can be modified to establish the local linear convergence of GD for matrix sensing (Zhu et al., 2021).*

## 4 RANDOM INITIALIZATION

Benign initialization has limited practical utility as it requires oracle information. This is particularly true in matrix sensing scenarios when $\boldsymbol{\Sigma}$ is only observed through random measurements (Stöger & Soltanolkotabi, 2021). Hence, researchers have begun to investigate random initialization. Note that by Theorem 2, the convergence analysis of GD reduces to studying how long it takes for the sequence to enter $\mathcal{R}$. Once the sequence enters $\mathcal{R}$, it will converge to the global minimum exponentially fast.

## 4.1 SMALL RANDOM INITIALIZATION

Existing works (except for the rank-one case) all consider the scenario of small random initialization. They assume $\boldsymbol{X}_0 = \varpi \boldsymbol{N}_0$, where $\boldsymbol{N}_0 \in \mathbb{R}^{d \times r}$ has independent $\mathcal{N}(0, \frac{1}{d})$ entries and $\varpi$ is very small. By the concentration results, the norm $\|\boldsymbol{X}_0\|$ is of order $\mathcal{O}(\varpi)$. When $\varpi$ is sufficiently small, the higher-order term $\boldsymbol{X}.\boldsymbol{X}.^\top \boldsymbol{X}.$ in (9) becomes negligible in the early stage. Consequently, in the early stage, the GD iteration behaves like a spectral method (or a power method):

$$\boldsymbol{X}_t \approx \boldsymbol{X}_{t-1} + \eta \boldsymbol{\Sigma} \boldsymbol{X}_{t-1}. \tag{11}$$

The eigenvectors associated with larger eigenvalues will be learned faster. Using the same $\boldsymbol{U}, \boldsymbol{J}$ in Section 3, we know $\sigma_r(\boldsymbol{U}_{t+1})/\sigma_r(\boldsymbol{U}_t)$ is greater than $\sigma_1(\boldsymbol{J}_{t+1})/\sigma_1(\boldsymbol{J}_t)$ for small $t$, meaning that the signal strength increases faster than the noise strength. As long as we pick a sufficiently small $\varpi$, we can show that after $\mathcal{O}(\log(d))$ rounds, $\sigma_r^2(\boldsymbol{U}_t)$ will rise above $\Delta/4$ while $\sigma_1(\boldsymbol{J}_t)$ remains negligible. This implies that the sequence $\boldsymbol{X}_t$ will enter the region $\mathcal{R}$ quickly, and combined with a local linear convergence result, Stöger & Soltanolkotabi (2021) demonstrate the linear global convergence of GD. In addition, Jin et al. (2023) reveal the incremental learning behavior of GD with a small $\varpi$.

These work typically require $\varpi = d^{-\iota(\kappa)}$ for some positive, increasing function $\iota(\cdot)$, where $\kappa = \lambda_1/\Delta \geq 1$ is the conditional number. For instance, Stöger & Soltanolkotabi (2021) require

$$\varpi \lesssim \min\{d^{-1/2}, d^{-3\kappa^2}\}. \tag{12}$$

Jin et al. (2023) require a even smaller $\varpi$. Such $\varpi$ decays to zero fast when $d$ increase or $\kappa$ increases.

## 4.2 LARGE RANDOM INITIALIZATION

In sharp contrast, practitioners often use large initialization with $\boldsymbol{X}_0 = \varpi \boldsymbol{N}_0$, where $\varpi$ is a constant **independent of** $d$. For this case, the arguments in Section 3 or Section 4.1 are insufficient for building effective theories. Specifically, the initial SNR is too low to use the arguments in Section 3. Also, the initial magnitude $\|\boldsymbol{X}_0\|$ is high, rendering the arguments in Section 4.1 unfeasible. To understand large initialization, we will give a delicate dynamic analysis, corresponding to Figure 1 and related discussions in the introduction.

To proceed, we first introduce some notations. Consider problem (1) with rank $r$ and assume without loss of generality that $\boldsymbol{\Sigma} = \operatorname{diag}(\lambda_1, \ldots, \lambda_d)$ is diagonal with decreasing diagonal elements. We assume the leading $r + 1$ eigenvalues of $\boldsymbol{\Sigma}$ are strictly decreasing, meaning that the eigengap $\Delta = \min_{i \leq r}\{\lambda_i - \lambda_{i+1}\}$ is positive. Let $\boldsymbol{X}_t$ be the GD sequence from (9) and $\boldsymbol{X}_0$ be the initial point. We define $\boldsymbol{u}_{k,t}$ as the $k$-th row of $\boldsymbol{X}_t$ and $\boldsymbol{K}_{k,t}$ as the $(k+1)$-to-$d$-th rows of $\boldsymbol{X}_t$. Their relationships to Figure 1 have been discussed in the introduction. Finally, to present our main theorem, we define the following quantities related to the GD trajectory.

- First, we define $t_{\text{init},1} = \min\{t \geq 0 \mid \boldsymbol{X}_t \in \mathcal{S}\}$ as the first time when $\boldsymbol{X}_t$ enters $\mathcal{S}$, where

$$\mathcal{S} = \{\boldsymbol{X} \in \mathbb{R}^{d \times r} \mid \sigma_1^2(\boldsymbol{X}) \leq 2\lambda_1, \sigma_1^2(\boldsymbol{K}_k) \leq \lambda_k - \frac{3\Delta}{4}, \forall k \leq r\}, \tag{13}$$

  and $\boldsymbol{K}_k$ stands for the $(k+1)$-to-$d$ rows of $\boldsymbol{X}$. Here $\mathcal{S}$ represents a set where the norms of $\boldsymbol{X}$ and $\boldsymbol{K}_k$ are suitably upper bounded.
- Next, we define two constants $t^*$ and $t^\sharp$ as follows:

$$t^* = \log\left(\frac{\Delta^2}{8\lambda_1^3 + 144r^2\lambda_1}\right)/\log(1 - \eta\Delta/6), \text{ and } t^\sharp = \log\left(\frac{\Delta}{4r}\right)/\log(1 - \eta\Delta/6). \tag{14}$$

- Finally, we define the following quantities successively until $T_{\boldsymbol{u}_r}$.
  - $T_{\boldsymbol{u}_k} = \min\{t \geq 0 \mid \sigma_1^2(\boldsymbol{u}_{k,t+t_{\text{init},k}}) \geq \Delta/2\}$. It characterizes the time when the $k$-th signal strength surpasses $\Delta/2$ since $t_{\text{init},k}$.
  - $t_k = t_{\text{init},k} + T_{\boldsymbol{u}_k} + t^*$.
  - $t_k^*$ is defined as the smallest integer such that

$$r(1 - \eta\Delta/6)^{t_k^*} \leq \sqrt{\frac{\Delta}{8}} \min\{\sigma_1(\boldsymbol{u}_{k+1,t_k+t_k^*}), \sqrt{\frac{\Delta}{2}}\}. \tag{15}$$

    $t_k^*$ characterizes the time when the $(k+1)$-th signal strength is no longer smaller than a geometrically decaying sequence.

- $t_{\text{init},k+1} = t_k + t_k^*$.

These quantities represent the durations of various stages of the GD convergence. In Theorem 6, we provide upper bounds for these quantities and characterize the behavior of GD in specific time. Our result is deterministic and applicable to the case of large random initialization.

**Assumption 4** *Assume $t_k^* < \infty$ for all $k \leq r$.*

**Assumption 5 (Transition Assumption)** *Assume $t_k^* = \mathcal{O}(\log(d))$ for all $k \leq r$.*

**Theorem 6** *Suppose $\eta \leq \frac{\Delta}{100\lambda_1^2}$, $\sigma_1(\boldsymbol{X}_0) \leq \frac{1}{\sqrt{3\eta}}$, $\boldsymbol{X}_t$ is the GD sequence, and Assumption 4 holds. Then we have*

1. *$t_{\text{init},1} = \mathcal{O}(\frac{1}{\eta\lambda_1} \log \frac{1}{6\eta\lambda_1}) + \mathcal{O}(\frac{1}{\eta\Delta} \log \frac{8\lambda_1}{\Delta})$, which is a small constant. Moreover, $\boldsymbol{X}_t \in \mathcal{S}$ for all $t \geq t_{\text{init},1}$. This property holds even without Assumption 4.*

2. *For all $k \leq r$, $t_k$ and $t_{\text{init},k}$ are finite. In addition, $T_{\boldsymbol{u}_k} = \mathcal{O}(\frac{4}{\eta\Delta} \log \frac{\Delta}{2\sigma_1^2(\boldsymbol{u}_{k,t_{\text{init},k}})})$.*

3. *For all $k \leq r$ and $t \geq t_{\text{init},k} + T_{\boldsymbol{u}_k}$, we have $\sigma_1^2(\boldsymbol{u}_{k,t}) \geq \frac{\Delta}{2}$.*

4. *For all $k < r$ and $t \geq t_k$, we have $\sigma_1(\boldsymbol{u}_{k,t}\boldsymbol{K}_{k,t}^\top) \leq (1 - \eta\Delta/6)^{t-t_k}$ and*

$$|p_{k,t}| \leq (2\lambda_1 + \frac{24r}{\eta\Delta}) \cdot (1 - \eta\Delta/8)^{t-t_k}, \tag{16}$$

   *where $p_{k,t} = \lambda_k - \sigma_1^2(\boldsymbol{u}_{k,t})$. This demonstrates the incremental learning of GD.*

5. *For all $t \geq t_{\mathcal{R}} := t_{\text{init},r} + T_{\boldsymbol{u}_r} + t^* + t^\sharp$, we have $\boldsymbol{X}_t \in \mathcal{R}$.*

6. *GD achieves $\epsilon$-accuracy, i.e., $\|\boldsymbol{\Sigma}_r - \boldsymbol{X}_t\boldsymbol{X}_t^\top\|_{\mathrm{F}} \leq \epsilon$, after $t_{\mathcal{R}} + \mathcal{O}(\frac{6}{\eta\Delta} \ln \frac{200r\lambda_1^3}{\eta\Delta^2\epsilon})$ iterations.*

7. *If Assumption 5 holds, then GD achieves $\epsilon$-accuracy in $\mathcal{O}(\log(d) + \log(1/\epsilon))$ iterations.*

Let us discuss about the assumptions and conclusions. First, we assume $\sigma_1(\boldsymbol{X}_0) \leq 1/\sqrt{3\eta}$. It holds with high probability when we use $\boldsymbol{X}_0 = \varpi \boldsymbol{N}_0$ with $\varpi \lesssim \frac{1}{\sqrt{\eta}}$ and the same $\boldsymbol{N}_0$ as before. This order $\frac{1}{\sqrt{\eta}}$ is optimal from the above, because the GD sequence may simply diverge when $\sigma_1(\boldsymbol{X}_0)$ is too large. For instance, consider $\boldsymbol{\Sigma} = \boldsymbol{0}$ and $\eta\sigma_1^2(\boldsymbol{X}_0) \geq 3$. By an inductive argument and the GD iteration (9), we can show that

$$\sigma_1(\boldsymbol{X}_{t+1}) \geq (\eta\sigma_1^2(\boldsymbol{X}_t) - 1) \cdot \sigma_1(\boldsymbol{X}_t) > 2\sigma_1(\boldsymbol{X}_t), \quad \forall t.$$

This implies that GD diverges in this scenario and justifies that our condition for $\varpi$ is rate optimal. The only possible improvement is a constant factor. In addition, we compare our requirements with (12) in the scenario of small initialization. Specifically, condition (12) decays to zero exponentially fast when $d$ increases, while our condition is independent of $d$.

Next, we emphasize that Assumption 4 almost surely holds if we use random initialization. This follows from the theory of Lee et al. (2016) and the landscape analysis of Zhu et al. (2021). Zhu et al. (2021) show that problem (1) only has strict saddle points and all local minima are global ones. Lee et al. (2016) prove that GD almost surely avoids strict saddle points. Combining these two results, we know GD converges to the global minimum for problem (1). This implies Assumption 4 because if it does not hold for some $k$, then $\sigma_1(\boldsymbol{u}_{k,t})$ will converge to zero and the GD sequence will converge to a saddle point. This case almost never happens. This proves the following proposition.

**Proposition 7** *Suppose $\eta \leq \frac{\Delta}{100\lambda_1^2}$. Then the following set*

$$\text{failure set} := \{\boldsymbol{X} \in \mathbb{R}^{d \times r} \mid \sigma_1(\boldsymbol{X}) \leq \frac{1}{\sqrt{3\eta}}, X_t \text{ is the GD sequence initialized with } \boldsymbol{X},$$

$$\text{and Assumption 4 does not hold for this sequence } \boldsymbol{X}_t.\}$$

*has measure zero.*

Third, Assumption 5 is a more advanced assumption because it upper bounds the quantity $t_k^*$. We call it a transition assumption because it allows us to transit the analysis from the $k$-th row to the $(k + 1)$-th row and it assumes the transition time is $\mathcal{O}(\log(d))$. With this assumption, we could obtain the seventh property in Theorem 6, that is, the fast global convergence of GD. Nevertheless, it is challenging to verify this assumption. In Section 5.1.3, we will give more discussions.

Fortunately, without Assumption 5, the first six properties in Theorem 6 still hold. These properties provide meaningful characterizations of the convergence of GD. Specifically, all quantities beyond $t_k^*$ are suitable upper bounded either by a constant or a logarithmic term. These bounds explain the fast convergence of GD in Figure 1 (to a certain degree). Moreover, the fourth property is noteworthy. It demonstrates that the $k$-th signal strength will converge linearly to the target value since the $t_k$-th step. This is independent of $t_j^*$ for all $j \geq k$. In other words, the $k$-th signal will converge fast to the target value independent of the behavior of the latter $((k + 1)$-to-$r$-th) signals. This explains the incremental learning phenomenon exhibited by GD.

## 5 PROOF SKETCH

In this section, we will provide a sketch of proof. We will start with rank-two matrix approximation and then extend it to general rank problems. The only difference between rank-two problem and general rank problems lies in how many rounds of inductive arguments are needed.

### 5.1 RANK-TWO MATRIX APPROXIMATION

To start with, we first show that when $\sigma_1(\boldsymbol{X}_0) \leq \frac{1}{\sqrt{3\eta}}$, the GD sequence will quickly enter the region $\mathcal{S}$ defined in (13), and the sequence will remain in $\mathcal{S}$ afterwards. This proves the first property in Theorem 6. Recall that $t_{\text{init},1} = \min\{t \geq 0 \mid \boldsymbol{X}_t \in \mathcal{S}\}$ and $\boldsymbol{X}_t$ is the GD sequence given by (9).

**Lemma 8** *Suppose* $\eta \leq \frac{1}{12\lambda_1}$ *and* $\sigma_1(\boldsymbol{X}_0) \leq \frac{1}{\sqrt{3\eta}}$. *Then* $\boldsymbol{X}_t \in \mathcal{S}$ *for all* $t \geq t_{\text{init},1}$, *where*

$$t_{\text{init},1} = \mathcal{O}\left(\frac{1}{\eta\lambda_1}\log\frac{1}{6\eta\lambda_1}\right) + \mathcal{O}\left(\frac{1}{\eta\Delta}\log\frac{8\lambda_1}{\Delta}\right).$$

Lemma 8 demonstrates that $\mathcal{S}$ is an absorbing set of GD, meaning that the sequence will remain in the set after its first entrance. This enables us to use the property $\boldsymbol{X}_t \in \mathcal{S}$ in subsequent analysis.

#### 5.1.1 $\sigma_1^2(\boldsymbol{u}_{1,t})$ INCREASES ABOVE $\Delta/2$

Our next step is to analyze the first row $\boldsymbol{u}_{1,t}$ of $\boldsymbol{X}_t$. This is in sharp contrast to the results in Section 3 and 4.1, where the first $r$ rows of $\boldsymbol{X}_t$ are analyzed together. Although using large initialization makes previous analysis infeasible, it is still manageable to examine only the first row of $\boldsymbol{X}_t$. In Lemma 9, we show that $\sigma_1^2(\boldsymbol{u}_{1,t})$ increases fast above $\Delta/2$, and it remains larger than that afterwards. This proves the second and third properties in Theorem 6 for $k = 1$. In addition, this aligns with the first stage of the GD dynamics as displayed in Figure 1.

**Lemma 9** *Suppose* $\eta \leq \frac{1}{12\lambda_1}$, $\sigma_1(\boldsymbol{X}_0) \leq \frac{1}{\sqrt{3\eta}}$, *and* $\sigma_1(\boldsymbol{u}_{1,t_{\text{init},1}}) > 0$. *Then* $\sigma_1^2(\boldsymbol{u}_{1,t}) \geq \frac{\Delta}{2}$ *for all* $t \geq t_{\text{init},1} + T_{\boldsymbol{u}_1}$, *where*

$$T_{\boldsymbol{u}_1} = \mathcal{O}\left(\frac{4}{\eta\Delta}\log\frac{\Delta}{2\sigma_1^2(\boldsymbol{u}_{1,t_{\text{init},1}})}\right).$$

#### 5.1.2 SNR CONVERGES LINEARLY TO INFINITY AND $\sigma_1^2(\boldsymbol{u}_{1,t})$ CONVERGES

Once $\sigma_1^2(\boldsymbol{u}_{1,t})$ exceeds $\frac{\Delta}{2}$, then by a technique similar to (7), we can show that the SNR $\frac{\sigma_1^2(\boldsymbol{u}_{1,t})}{\sigma_1(\boldsymbol{u}_{1,t}\boldsymbol{K}_{1,t}^\top)}$ converges linearly to infinity, where $\boldsymbol{K}_{1,t}$ is the 2-to-$d$-th rows of $\boldsymbol{X}_t$. Since $\sigma_1^2(\boldsymbol{u}_{1,t})$ belong to the interval $[\Delta/2, 2\lambda_1]$ by Lemma 8 and 9, we can show that the noise strength $\sigma_1(\boldsymbol{u}_{1,t}\boldsymbol{K}_{1,t}^\top)$ diminishes to zero fast. In particular, when $\boldsymbol{u}_{1,t}\boldsymbol{K}_{1,t}^\top = \boldsymbol{0}$, the dynamics of $\boldsymbol{u}_{1,t}$ becomes

$$\boldsymbol{u}_{1,t+1} = \boldsymbol{u}_{1,t} + \eta\lambda_1\boldsymbol{u}_{1,t} - \eta\sigma_1^2(\boldsymbol{u}_{1,t})\boldsymbol{u}_{1,t}.$$

This update rule implies the fast convergence of $\sigma_1^2(\boldsymbol{u}_{1,t})$ to $\lambda_1$. Generally, when the term $\boldsymbol{u}_{1,t}\boldsymbol{K}_{1,t}^\top$ is close to zero, the dynamics of $\boldsymbol{u}_{1,t}$ will mimic the above iteration. Following this, we can establish the fast convergence of $\sigma_1^2(\boldsymbol{u}_{1,t})$ to $\lambda_1$. These results are established in Lemma 10. This relates to property 4 in Theorem 6, and elucidates the second stage of the GD dynamics as depicted in Figure 1.

**Lemma 10** *Suppose $\eta \leq \frac{\Delta}{100\lambda_1^2}$, $\sigma_1(\boldsymbol{X}_0) \leq \frac{1}{\sqrt{3\eta}}$, and $\sigma_1(\boldsymbol{u}_{1,0}) > 0$. Then for all $t \geq t_1$, we have*

$$\sigma_1(\boldsymbol{u}_{1,t}\boldsymbol{K}_{1,t}^\top) \leq (1 - \eta\Delta/6)^{t-t_1}$$

*where $t_1 = t_{\mathrm{init},1} + T_{\boldsymbol{u}_1} + t^*$, $T_{\boldsymbol{u}_1}$ is given in Lemma 9, and $t^*$ is a constant defined in (14). In addition, let $p_{1,t} = \lambda_1 - \sigma_1^2(\boldsymbol{u}_{1,t})$ be the error term. Then for all $t \geq t_1$, we have*

$$|p_{1,t}| \leq (2\lambda_1 + 24\frac{24r}{\eta\Delta}) \cdot (1 - \eta\Delta/8)^{t-t_1}.$$

### 5.1.3 Transition Assumption and Induction

Lemma 10 shows that the magnitude $\sigma_1(\boldsymbol{u}_{1,t}\boldsymbol{K}_{1,t}^\top)$ diminishes linearly to zero. This motivates us to decouple the original matrix factorization problem into two sub-problems. For the first sub-problem, we study the convergence of the first row of $\boldsymbol{X}_t$, which has been presented in previous section. In the second sub-problem, we examine $\boldsymbol{K}_{1,t}$, the 2-to-$d$-th rows of $\boldsymbol{X}_t$. Such decoupling is exact when $\boldsymbol{u}_{1,t}\boldsymbol{K}_{1,t}^\top = 0$; under this condition, the update rule of $\boldsymbol{K}_{1,t}$ becomes

$$\boldsymbol{K}_{1,t} = \boldsymbol{K}_{1,t-1} + \eta(\boldsymbol{\Gamma}_1 - \boldsymbol{K}_{1,t-1}\boldsymbol{K}_{1,t-1}^\top)\boldsymbol{K}_{1,t-1},$$

where $\boldsymbol{\Gamma}_1 = \mathrm{diag}(\lambda_2, \ldots, \lambda_d)$. This is congruent with the GD update rule of $\boldsymbol{X}_t$ as in (9), and hence an inductive argument could be applied.

Generally, when the noise term $\sigma_1(\boldsymbol{u}_{1,t}\boldsymbol{K}_{1,t}^\top)$ only decreases fast but does not reach zero, one should check whether $\boldsymbol{u}_{1,t}\boldsymbol{K}_{1,t}^\top$ is negligible (in the analysis of $\boldsymbol{u}_{2,t}$). Specifically, if $\sigma_1(\boldsymbol{u}_{2,t})$ is not always decreasing at the same speed as $\sigma_1(\boldsymbol{u}_{1,t}\boldsymbol{K}_{1,t}^\top)$, then we can apply the above inductive argument. To formulate this intuition, we introduce a variable $t_1^*$. It is defined as the smallest integer such that

$$r(1 - \eta\Delta/6)^{t_1^*} \leq \sqrt{\frac{\Delta}{8}}\min\{\sigma_1(\boldsymbol{u}_{2,t_1+t_1^*}), \sqrt{\frac{\Delta}{2}}\}, \tag{17}$$

where $t_1$ is defined in Lemma 10. Recall that for all $t \geq t_1$, $\sigma_1(\boldsymbol{u}_{1,t}\boldsymbol{K}_{1,t}^\top) \leq (1 - \eta\Delta/6)^{t-t_1}$. Hence, (17) essentially compares the second signal strength $\sigma_1(\boldsymbol{u}_{2,\cdot})$ with an upper bound on the noise term $\sigma_1(\boldsymbol{u}_{1,t}\boldsymbol{K}_{1,t}^\top)$. It turns out that the noise term is negligible when (17) holds. In particular, a similar result as Lemma 9 can be established for the second signal $\sigma_1(\boldsymbol{u}_{2,\cdot})$, leading to Lemma 11. It is also related to the second and third property in Theorem 6 (for $k = 2$).

**Lemma 11** *Suppose conditions of Lemma 10 holds. Let $t_{\mathrm{init},2} = t_1 + t_1^*$, where $t_1$ is given by Lemma 10 and $t_1^*$ is given by (17). Suppose $t_1^* < \infty$. Then $\sigma_1^2(\boldsymbol{u}_{2,t}) \geq \frac{\Delta}{2}$ for all $t \geq t_{\mathrm{init},2} + T_{\boldsymbol{u}_2}$, where*

$$T_{\boldsymbol{u}_2} = \mathcal{O}\left(\frac{4}{\eta\Delta}\log\frac{\Delta}{2\sigma_1^2(\boldsymbol{u}_{2,t_{\mathrm{init},2}})}\right).$$

In Lemma 11, we assume $t_1^* < \infty$, which relates to Assumption 4. If we assume $t_1^* = \mathcal{O}(\log(d))$ as in Assumption 5, then we can show that $T_{\boldsymbol{u}_2} = \mathcal{O}(\log(d))$ as well. While we have not theoretically characterized the quantity $t_1^*$, our theories are still insightful in the following sense.

- First, the term $\sigma_1(\boldsymbol{u}_{1,t}\boldsymbol{K}_{1,t}^\top)$ is shown to decay to zero linearly fast while $\sigma_1^2(\boldsymbol{u}_{2,t})$ does not seem to possess similar theories. Hence, we may expect that the time point $t_1^*$ is not large.

- Second, $t_1^*$ characterizes the time when the GD sequence escapes from the saddle points[2]. This time is inevitable for the GD sequence converging to the global minima. Even we do not provide an upper bound on $t_1^*$, we know the convergence behavior of GD during this time. Notably, during this time, both $\sigma_1(\boldsymbol{u}_{1,t}\boldsymbol{K}_{1,t}^\top)$ and $\sigma_1^2(\boldsymbol{u}_{2,t})$ converge to zero fast.

---

[2]Any stationary point with $\boldsymbol{u}_2 = 0$ is a saddle point. Hence, if the GD sequence $\boldsymbol{X}_t$ converges with $t_1^* = \infty$, then it must converge to a saddle point.

- Thirdly, during the time $t_1$-$(t_1+t_1^*)$, while $\sigma_1^2(\boldsymbol{u}_{2,t})$ converges to zero fast, the first signal $\sigma_1^2(\boldsymbol{u}_{1,t})$ still converges to $\lambda_1$, as shown in Lemma 10. This means the convergence of the first signal is not affected by the behaviors of the rest signals, which supports the incremental learning phenomenon – *leading signals first converge even when the rest are stuck by saddle points*.

- Finally, the time $t_1$ to $t_1 + t_1^*$ aligns with the third stage of the GD dynamics as displayed in Figure 1. The experiment shows that the time $t_1^*$ is not too long.

Despite these arguments, there is still a need to examine the duration $t_1^*$ in the future research, which might involve investigating specific initialization mechanisms.

### 5.1.4 FINAL CONVERGENCE

Since $r = 2$, the analysis of previous three stages implies that both the first signal strengths are larger than $\Delta/2$, and the related noise components are geometrically decaying. A simple verification shows that the GD sequence $\boldsymbol{X}_t$ will quickly enter the region $\mathcal{R}$, which is defined in (10). Then by the local linear convergence of GD in Theorem 2, we shall complete the characterization of the GD sequence's convergence to the global minima. This final stage aligns with the fourth stage of the GD dynamics as illustrated in Figure 1.

### 5.2 GENERAL RANK MATRIX APPROXIMATION

It is direct to extend rank-two matrix approximation to general rank case. The key point is to repeat the inductive arguments for $(r-1)$ rather than one times. Similar to the rank-two case, we will now successively show that $\sigma_1^2(\boldsymbol{u}_{k,t})$ surpasses $\Delta/2$ and $\sigma_1(\boldsymbol{u}_{k,t}\boldsymbol{K}_{k,t}^\top)$ diminishes linearly to zero for all $k \leq r$. Moreover, we will show that $\sigma_1^2(\boldsymbol{u}_{k,t})$ converges to $\lambda_k$ after certain iterations. Once the first $r$ rows of $\boldsymbol{X}_t$ are all analyzed, we can show that the sequence $\boldsymbol{X}_t$ quickly enters the region $\mathcal{R}$ defined in (10). By invoking the local linear convergence theorem, we will conclude the proof.

Our analysis consistently uses the SNR argument, where the choices of SNRs vary across different contexts. Specifically,

- When we analyze the $k$-th signal strength in Theorem 6, we will analyze the SNR $\frac{\sigma_1^2(\boldsymbol{u}_{k,t})}{\sigma_1(\boldsymbol{u}_{k,t}\boldsymbol{K}_{k,t}^\top)}$. This will prove both the diminishing of $\sigma_1(\boldsymbol{u}_{k,t}\boldsymbol{K}_{k,t}^\top)$ and the convergence of $\sigma_1^2(\boldsymbol{u}_{k,t})$ to $\lambda_k$.

- When we analyze the local linear convergence in Theorem 2, we will take the SNR as $\frac{\sigma_r^2(\boldsymbol{U}_t)}{\sigma_1^2(\boldsymbol{J}_t)}$, where $\boldsymbol{U}, \boldsymbol{J}$ are defined in Section 3. Such analysis will prove the linear convergence of $\boldsymbol{J}$ to zero.

## 6 CONCLUDING REMARKS

This paper presents a comprehensive analysis of the trajectory of GD in addressing matrix factorization issues, emphasizing particularly on instances with large initialization. The analysis employs both a SNR argument and an induction argument to bolster the investigation's depth and insight. Our finding is that even with large initialization, GD may still exhibit an incremental learning phenomenon. Also, the main challenging convergence issue is to escape from the saddle points. We hope our findings can inspire other researchers in related fields.

There are several limitations within this paper, bringing future research opportunities.

- First, we do not upper bound the time $t_k^*$ defined in (14). Hence, it is of interest to give an effective upper bound. Also, one may examine this point to see if negative results can be established.

- Second, our paper requires a strictly decreasing top eigenvalues. Extending to more general matrices may need additional studies.

- Third, our analysis focuses on the simplest matrix factorization setting. It is intriguing to study similar results in other settings, such as matrix sensing, where $\boldsymbol{\Sigma}$ is only accessible via linear measurements. Our delicate dynamic analysis is sensitive to the noise introduced by the measurement mechanism. Hence, new theoretical tools are needed.

- Fourth, it is interesting to examine GD in solving deep matrix factorization. It is unknown how large initialization affects the GD trajectory in that case.

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
