APPENDIX

## A    PROOF OF THEOREM 2

Our proof of Theorem 2 consists of three steps.

- First, we show that $\mathcal{R}$ is an absorbing region for GD. Here a set is regarded as an absorbing set if the GD sequence remains within the set after its first entrance.
- Next, we show that $\sigma_1(\boldsymbol{J}_t)$ converges to zero at a linear rate, employing an SNR argument.
- Finally, we establish the linear convergence to the global minima.

Before diving deeper, we first write down the update rules for $\boldsymbol{U}_t$ and $\boldsymbol{J}_t$. By (9), we have

$$\boldsymbol{U}_{t+1} = \boldsymbol{U}_t + \eta\boldsymbol{\Lambda}_r\boldsymbol{U}_t - \eta\boldsymbol{U}_t\boldsymbol{X}_t^\top\boldsymbol{X}_t, \qquad (18)$$

$$\boldsymbol{J}_{t+1} = \boldsymbol{J}_t + \eta\boldsymbol{\Lambda}_{\mathrm{res}}\boldsymbol{J}_t - \eta\boldsymbol{J}_t\boldsymbol{X}_t^\top\boldsymbol{X}_t, \qquad (19)$$

where $\boldsymbol{\Lambda}_r = \mathrm{diag}(\lambda_1, \ldots, \lambda_r)$ and $\boldsymbol{\Lambda}_{\mathrm{res}} = \mathrm{diag}(\lambda_{r+1}, \ldots, \lambda_d)$. Note that $\boldsymbol{\Sigma}_r = \mathrm{diag}(\boldsymbol{\Lambda}_r, \boldsymbol{0})$.

### A.1    THE GD SEQUENCE REMAINS IN $\mathcal{R}$

Lemma 12 shows that $\mathcal{R}$ is an absorbing region for GD.

**Lemma 12** *Suppose $\eta \leq \frac{\Delta^2}{36\lambda_1^3}$ and $\boldsymbol{X}_t \in \mathcal{R}$. Then $\boldsymbol{X}_{t'} \in \mathcal{R}$ for all $t' \geq t$.*

**Proof** This lemma is proved by induction. Suppose $\boldsymbol{X}_t \in \mathcal{R}$.

- By Lemma 13 and $\sigma_1^2(\boldsymbol{X}_t) \leq 2\lambda_1$, we get $\sigma_1^2(\boldsymbol{X}_{t+1}) \leq 2\lambda_1$.
- By Lemma 14, $\sigma_1^2(\boldsymbol{X}_t) \leq 2\lambda_1$, and $\sigma_1^2(\boldsymbol{J}_t) \leq \lambda_r - \Delta/2$, we get $\sigma_1^2(\boldsymbol{J}_{t+1}) \leq \lambda_r - \Delta/2$.
- By Lemma 15 and $\boldsymbol{X}_t \in \mathcal{R}$, we get $\sigma_r^2(\boldsymbol{U}_{t+1}) \geq \Delta/4$ and thus $\boldsymbol{X}_{t+1} \in \mathcal{R}$.

By induction, we conclude that $\boldsymbol{X}_{t'} \in \mathcal{R}$ for all $t' \geq t$. ∎

#### A.1.1    TECHNICAL LEMMAS

In this section, we summarize technical lemmas used in the proof of Lemma 12.

Lemma 13 delineates the first category of absorbing sets for GD, denoted as

$$\mathcal{S}_1 = \{\boldsymbol{X} \in \mathbb{R}^{d \times r} \mid \sigma_1(\boldsymbol{X}) \leq a\},$$

valid for any $a \in [\sqrt{\lambda_1}, 1/\sqrt{3\eta}]$.

**Lemma 13** *Suppose $\eta \leq \frac{1}{3\lambda_1}$ and $a \in [\sqrt{\lambda_1}, 1/\sqrt{3\eta}]$. If $\sigma_1(\boldsymbol{X}_t) \leq a$, then $\sigma_1(\boldsymbol{X}_{t'}) \leq a, \forall t' \geq t$.*

**Proof** Lemma 16 states that if $\sigma_1(\boldsymbol{X}_t) \leq 1/\sqrt{3\eta}$, then the following inequality holds

$$\sigma_1(\boldsymbol{X}_{t+1}) \leq (1 + \eta\lambda_1 - \eta\sigma_1^2(\boldsymbol{X}_t)) \cdot \sigma_1(\boldsymbol{X}_t).$$

- If $\sqrt{\lambda_1} \leq \sigma_1(\boldsymbol{X}_t) \leq a$, the above inequality implies that $\sigma_1(\boldsymbol{X}_{t+1}) \leq \sigma_1(\boldsymbol{X}_t) \leq a$.
- If $\sigma_1(\boldsymbol{X}_t) \leq \sqrt{\lambda_1} \leq a$, it follows that

$$\sigma_1(\boldsymbol{X}_{t+1}) \leq (1 + \eta\lambda_1 - \eta\lambda_1)\sqrt{\lambda_1} \leq a.$$

This uses the fact that $g_1(s) = (1 + \eta\lambda_1 - \eta s^2)s$ is increasing on $[0, 1/\sqrt{3\eta}]$.

By induction, we have $\sigma_1(\boldsymbol{X}_{t'}) \le a$ for all $t' \ge t$. ∎

Lemma 14 demonstrates that if $\sigma_1(\boldsymbol{X}_t) \le \sqrt{2\lambda_1}$, $\sigma_1^2(\boldsymbol{J}_t) \le a$, and $a \ge \lambda_{r+1}$, then $\sigma_1^2(\boldsymbol{J}_{t+1}) \le a$. Combining with Lemma 13, it implies that

$$\mathcal{S}_2 = \{ \boldsymbol{X} = \begin{pmatrix} \boldsymbol{U} \\ \boldsymbol{J} \end{pmatrix} \in \mathbb{R}^{d \times r} \mid \sigma_1(\boldsymbol{X}) \le \sqrt{2\lambda_1}, \sigma_1^2(\boldsymbol{J}) \le a \}$$

is an absorbing set for GD, provided that $a \ge \lambda_{r+1}$ and $\eta \le \frac{1}{12\lambda_1}$. Here $\boldsymbol{U}$ and $\boldsymbol{J}$ are the top $r$ rows and the $(r+1)$-to-$d$-th rows of $\boldsymbol{X}$ respectively.

**Lemma 14** *Suppose* $\eta \le \frac{1}{12\lambda_1}$, $\sigma_1^2(\boldsymbol{X}_t) \le 2\lambda_1$, *and* $a \ge \lambda_{r+1}$. *If* $\sigma_1^2(\boldsymbol{J}_t) \le a$, *then* $\sigma_1^2(\boldsymbol{J}_{t+1}) \le a$.

**Proof** By Lemma 17, we have

$$\sigma_1(\boldsymbol{J}_{t+1}) \le (1 + \eta(\lambda_{r+1} - \sigma_1^2(\boldsymbol{J}_t))) \cdot \sigma_1(\boldsymbol{J}_t).$$

- If $\lambda_{r+1} < \sigma_1^2(\boldsymbol{J}_t) \le a$, then it follows that $\sigma_1^2(\boldsymbol{J}_{t+1}) \le \sigma_1^2(\boldsymbol{J}_t) \le a$.
- If $\sigma_1^2(\boldsymbol{J}_t) \le \lambda_{r+1} \le a$, then

$$\sigma_1^2(\boldsymbol{J}_{t+1}) \le (1 + \eta(\lambda_{r+1} - \lambda_{r+1}))^2 \lambda_{r+1} \le a.$$

  This uses the observation that $g_2(s) = (1 + \eta(\lambda_{r+1} - s^2))s$ is increasing on $[0, 1/\sqrt{3\eta}]$.

This concludes the proof. ∎

Lemma 15 is the last piece needed to show that region $\mathcal{R}$ is an absorbing set for GD.

**Lemma 15** *Suppose* $\eta \le \frac{\Delta^2}{32\lambda_1^3}$, $\sigma_1(\boldsymbol{X}_t) \le \sqrt{2\lambda_1}$, *and* $\sigma_1^2(\boldsymbol{J}_t) \le \lambda_r - \Delta/2$. *If* $\sigma_r^2(\boldsymbol{U}_t) \ge \Delta/4$, *then* $\sigma_r^2(\boldsymbol{U}_{t+1}) \ge \Delta/4$.

**Proof** Since $\eta \le \frac{1}{32\lambda_1}$ and $\sigma_1^2(\boldsymbol{J}_t) \le \lambda_r - \Delta/2$, by Lemma 18, we have

$$\sigma_r^2(\boldsymbol{U}_{t+1}) \ge (1 + \eta\Delta - 2\eta\sigma_r^2(\boldsymbol{U}_t)) \cdot \sigma_r^2(\boldsymbol{U}_t) - 4\eta^2\lambda_1^3.$$

Since $g_3(s) = (1 + \eta\Delta - 2\eta s)s$ is increasing on $(-\infty, \frac{1}{4\eta}]$ and $\frac{\Delta}{4} \le \sigma_r^2(\boldsymbol{U}_t) \le 2\lambda_1 \le \frac{1}{4\eta}$, we have

$$\sigma_r^2(\boldsymbol{U}_{t+1}) \ge (1 + \frac{\eta\Delta}{2}) \cdot \frac{\Delta}{4} - 4\eta^2\lambda_1^3 \ge \frac{\Delta}{4},$$

where the last inequality uses $\eta \le \frac{\Delta^2}{32\lambda_1^3}$. ∎

The following lemmas give certain singular value analysis that are used in prior lemmas and subsequent analysis. Lemma 16 establishes an upper bound for $\sigma_1(\boldsymbol{X}_{t+1})$.

**Lemma 16** *If* $\sigma_1(\boldsymbol{X}_t) \le 1/\sqrt{3\eta}$, *then we have*

$$\sigma_1(\boldsymbol{X}_{t+1}) \le (1 + \eta\lambda_1 - \eta\sigma_1^2(\boldsymbol{X}_t)) \cdot \sigma_1(\boldsymbol{X}_t).$$

**Proof** By the singular value inequality and (9),

$$\sigma_1(\boldsymbol{X}_{t+1}) \le \sigma_1(\boldsymbol{X}_t(\boldsymbol{I}_r - \eta\boldsymbol{X}_t^\top\boldsymbol{X}_t)) + \eta\sigma_1(\boldsymbol{\Sigma}\boldsymbol{X}_t)$$
$$\le \sigma_1(\boldsymbol{X}_t(\boldsymbol{I}_r - \eta\boldsymbol{X}_t^\top\boldsymbol{X}_t)) + \eta\lambda_1\sigma_1(\boldsymbol{X}_t), \tag{20}$$

where we use $\sigma_1(\boldsymbol{\Sigma}) = \lambda_1$. Observe that all $r$ singular values of $\boldsymbol{X}_t(\boldsymbol{I}_r - \eta\boldsymbol{X}_t^\top\boldsymbol{X}_t)$ are given by

$$(1 - \eta\sigma_i^2(\boldsymbol{X}_t)) \cdot \sigma_i(\boldsymbol{X}_t), \ i = 1, \ldots, r,$$

since $\eta\sigma_1^2(\boldsymbol{X}_t) \le 1$. The function $g_4(s) = (1 - \eta s^2)s$ is increasing on $[0, 1/\sqrt{3\eta}]$. Hence, the fact $0 \le \sigma_i(\boldsymbol{X}_t) \le \sigma_1(\boldsymbol{X}_t) \le 1/\sqrt{3\eta}$ implies that

$$\sigma_1(\boldsymbol{X}_t(\boldsymbol{I}_r - \eta\boldsymbol{X}_t^\top\boldsymbol{X}_t)) = (1 - \eta\sigma_1^2(\boldsymbol{X}_t)) \cdot \sigma_1(\boldsymbol{X}_t).$$

Substituting this equality into (20), we conclude the proof. ∎

Lemma 17 gives an upper bound for $\sigma_1(\boldsymbol{J}_{t+1})$.

**Lemma 17** *Suppose $\eta \leq \frac{1}{12\lambda_1}$ and $\sigma_1(\boldsymbol{X}_t) \leq \sqrt{2\lambda_1}$, then we have*

$$\sigma_1(\boldsymbol{J}_{t+1}) \leq (1 + \eta(\lambda_{r+1} - \sigma_1^2(\boldsymbol{J}_t) - \sigma_r^2(\boldsymbol{U}_t))) \cdot \sigma_1(\boldsymbol{J}_t).$$

**Proof** The update rule (19) of $\boldsymbol{J}_{t+1}$ can be decomposed as follows:

$$\boldsymbol{J}_{t+1} = \underbrace{\frac{1}{2}\boldsymbol{J}_t - \eta\boldsymbol{J}_t\boldsymbol{J}_t^\top\boldsymbol{J}_t}_{\boldsymbol{B}} + \underbrace{(\frac{1}{4}\boldsymbol{I}_{d-r} + \eta\boldsymbol{\Lambda}_{\text{res}})\boldsymbol{J}_t}_{\boldsymbol{C}} + \underbrace{\boldsymbol{J}_t(\frac{1}{4}\boldsymbol{I}_r - \eta\boldsymbol{U}_t^\top\boldsymbol{U}_t)}_{\boldsymbol{D}}.$$

By the singular value inequality,

$$\sigma_1(\boldsymbol{J}_{t+1}) \leq \sigma_1(\boldsymbol{B}) + \sigma_1(\boldsymbol{C}) + \sigma_1(\boldsymbol{D}).$$

Observe that all singular values of $\boldsymbol{B}$ are given by

$$\sigma_i(\boldsymbol{J}_t)/2 - \eta\sigma_i^3(\boldsymbol{J}_t), \quad i = 1, \ldots, d-r.$$

Since $g_5(s) = s/2 - \eta s^3$ is increasing on $[0, 1/\sqrt{6\eta}]$, the condition $\sigma_i(\boldsymbol{J}_t) \leq \sigma_1(\boldsymbol{J}_t) \leq \sqrt{2\lambda_1} \leq 1/\sqrt{6\eta}$ implies that

$$\sigma_1(\boldsymbol{B}) = \sigma_1(\boldsymbol{J}_t)/2 - \eta\sigma_1^3(\boldsymbol{J}_t).$$

For the second term $\boldsymbol{C}$, it follows from the singular value inequality that

$$\sigma_1(\boldsymbol{C}) \leq \sigma_1(\frac{1}{4}\boldsymbol{I}_{d-r} + \eta\boldsymbol{\Lambda}_{\text{res}})\sigma_1(\boldsymbol{J}_t) \leq (1/4 + \eta\lambda_{r+1})\sigma_1(\boldsymbol{J}_t),$$

where the second inequality uses $\eta\sigma_1(\boldsymbol{\Lambda}_{\text{res}}) \leq \eta\lambda_1 \leq 1/4$. For the third term $\boldsymbol{D}$, since $\eta\sigma_1^2(\boldsymbol{U}_t) \leq 2\eta\lambda_1 \leq 1/4$, we have

$$\sigma_1(\boldsymbol{D}) \leq (1/4 - \eta\sigma_r^2(\boldsymbol{U}_t))\sigma_1(\boldsymbol{J}_t).$$

Finally, we conclude the proof by combining the analysis of $\boldsymbol{B}, \boldsymbol{C}$, and $\boldsymbol{D}$. $\blacksquare$

Lemma 18 provides an lower bound for $\sigma_r^2(\boldsymbol{U}_{t+1})$.

**Lemma 18** *Suppose $\eta \leq \frac{1}{32\lambda_1}$ and $\sigma_1(\boldsymbol{X}_t) \leq \sqrt{2\lambda_1}$, then we have*

$$\sigma_r^2(\boldsymbol{U}_{t+1}) \geq (1 + 2\eta(\lambda_r - \sigma_1^2(\boldsymbol{J}_t) - \sigma_r^2(\boldsymbol{U}_t))) \cdot \sigma_r^2(\boldsymbol{U}_t) - 4\eta^2\lambda_1^3.$$

**Proof** Substituting the update rule (18) of $\boldsymbol{U}_{t+1}$ into $\boldsymbol{U}_{t+1}\boldsymbol{U}_{t+1}^\top$, we get

$$\begin{aligned}
\boldsymbol{U}_{t+1}\boldsymbol{U}_{t+1}^\top &= (\boldsymbol{U}_t - \eta\boldsymbol{U}_t\boldsymbol{X}_t^\top\boldsymbol{X}_t + \eta\boldsymbol{\Lambda}_r\boldsymbol{U}_t) \cdot (\boldsymbol{U}_t - \eta\boldsymbol{U}_t\boldsymbol{X}_t^\top\boldsymbol{X}_t + \eta\boldsymbol{\Lambda}_r\boldsymbol{U}_t)^\top \\
&= \boldsymbol{B} + \boldsymbol{C} - \eta^2\boldsymbol{R}_1 + \eta^2\boldsymbol{R}
\end{aligned}$$

where

$$\begin{aligned}
\boldsymbol{B} &= \boldsymbol{U}_t(\frac{1}{2}\boldsymbol{I}_r - 2\eta\boldsymbol{X}_t^\top\boldsymbol{X}_t)\boldsymbol{U}_t^\top, \\
\boldsymbol{C} &= (\frac{1}{\sqrt{2}}\boldsymbol{I}_r + \sqrt{2}\eta\boldsymbol{\Lambda}_r)\boldsymbol{U}_t\boldsymbol{U}_t^\top(\frac{1}{\sqrt{2}}\boldsymbol{I}_r + \sqrt{2}\eta\boldsymbol{\Lambda}_r), \\
\boldsymbol{R}_1 &= 2\boldsymbol{\Lambda}_r\boldsymbol{U}_t\boldsymbol{U}_t^\top\boldsymbol{\Lambda}_r, \\
\boldsymbol{R} &= (\boldsymbol{\Lambda}_r\boldsymbol{U}_t - \boldsymbol{U}_t\boldsymbol{X}_t^\top\boldsymbol{X}_t)(\boldsymbol{\Lambda}_r\boldsymbol{U}_t - \boldsymbol{U}_t\boldsymbol{X}_t^\top\boldsymbol{X}_t)^\top.
\end{aligned}$$

Here $\boldsymbol{B}$ is positive semi-definite (PSD) since $2\eta\sigma_1^2(\boldsymbol{X}_t) \leq 4\eta\lambda_1 \leq 1/2$ and $\boldsymbol{C}, \boldsymbol{R}_1, \boldsymbol{R}$ are all PSD. By the eigenvalue inequality and the equivalence between eigenvalues and singular values of a PSD matrix, we have

$$\begin{aligned}
\sigma_r^2(\boldsymbol{U}_{t+1}) &\geq \sigma_r(\boldsymbol{B}) + \sigma_r(\boldsymbol{C}) - \eta^2\sigma_1(\boldsymbol{R}_1) + \eta^2\sigma_r(\boldsymbol{R}) \\
&\geq \sigma_r(\boldsymbol{B}) + \sigma_r(\boldsymbol{C}) - \eta^2\sigma_1(\boldsymbol{R}_1). \tag{21}
\end{aligned}$$

For the first term $B$, we decompose it into two terms:

$$B = \underbrace{U_t((\frac{1}{2} - 2\eta\sigma_1^2(J_t)) \cdot I_r - 2\eta U_t^\top U_t)U_t^\top}_{B_1} + 2\eta \cdot \underbrace{U_t(\sigma_1^2(J_t) \cdot I_r - J_t^\top J_t)U_t^\top}_{B_2}.$$

The inequality $2\eta(\sigma_1^2(J_t) + \sigma_1^2(U_t)) \le 8\eta\lambda_1 \le 1/2$ implies that $B_1$ is PSD. Since $B_2$ is also PSD, we have $\sigma_r(B) \ge \sigma_r(B_1)$. To determine $\sigma_r(B_1)$, we write the singular values of $B_1$ as

$$(\frac{1}{2} - 2\eta\sigma_1^2(J_t)) \cdot \sigma_i^2(U_t) - 2\eta\sigma_i^4(U_t), \ i = 1, \ldots, r.$$

Since $1/2 - 2\eta\sigma_1^2(J_t) \ge 1/4$, the function $g_6(s) = (1/2 - 2\eta\sigma_1^2(J_t))s - 2\eta s^2$ is increasing on $(-\infty, \frac{1}{16\eta}]$. Then the inequality $\sigma_i^2(U_t) \le \sigma_1^2(U_t) \le 2\lambda_1 \le \frac{1}{16\eta}$ implies that

$$\sigma_r(B_1) = (\frac{1}{2} - 2\eta(\sigma_1^2(J_t) + \sigma_r^2(U_t))) \cdot \sigma_r^2(U_t).$$

For the second term $C$, we have

$$\sigma_r(C) \ge \sigma_r^2(\frac{1}{\sqrt{2}}I_r + \sqrt{2}\eta\Lambda_r)\sigma_r^2(U_t) \ge (\frac{1}{2} + 2\eta\lambda_r)\sigma_r^2(U_t).$$

For the third term $R_1$, since $\sigma_1^2(X_t) \le 2\lambda_1$, we have

$$\sigma_1(R_1) \le 4\lambda_1^3.$$

Finally, substituting the analysis of $B, C, R_1$ into (21) gives the desired result. ∎

## A.2 $\sigma_1(J_t)$ CONVERGES TO ZERO LINEARLY VIA AN SNR ARGUMENT

Lemma 19 shows that if $X_0 \in \mathcal{R}$, then $\sigma_1(J_t)$ will diminish to zero at a geometric rate. A key step of the analysis is to examine the SNR $\frac{\sigma_r^2(U_t)}{\sigma_1^2(J_t)}$. Our analysis extends the rank-one case in Section 2 to a general rank scenario.

**Lemma 19** *Suppose $\eta \le \Delta^2/(32\lambda_1^3)$ and $X_0 \in \mathcal{R}$. Then, for all $t \ge 0$, we have*

$$\frac{\sigma_1^2(J_{t+1})}{\sigma_r^2(U_{t+1})} \le (1 - \eta\Delta/3) \cdot \frac{\sigma_1^2(J_t)}{\sigma_r^2(U_t)}.$$

*Hence, $\sigma_1^2(J_t) \le 8\lambda_1^2(1 - \eta\Delta/3)^t/\Delta$ for all $t$ and $\sigma_1^2(J_t) < \epsilon$ after*

$$T_J^\epsilon = \mathcal{O}\left(\frac{3}{\eta\Delta} \log \frac{8\lambda_1^2}{\epsilon\Delta}\right) \ iterations.$$

**Proof** By Lemma 12, we have $X_t \in \mathcal{R}$ for all $t \ge 0$. Then by Lemma 17,

$$\sigma_1^2(J_{t+1}) \le (1 + 2\eta(\lambda_{r+1} - \sigma_1^2(J_t) - \sigma_r^2(U_t)) + 16\eta^2\lambda_1^2) \cdot \sigma_1^2(J_t)$$
$$\le (1 - \eta\Delta/2 + 2\eta(\lambda_r - \Delta/2 - \sigma_1^2(J_t) - \sigma_r^2(U_t))) \cdot \sigma_1^2(J_t),$$

where the second inequality follows from $\eta \le \frac{\Delta}{32\lambda_1^2}$. By Lemma 18,

$$\sigma_r^2(U_{t+1}) \ge (1 + \eta\Delta + 2\eta(\lambda_r - \Delta/2 - \sigma_1^2(J_t) - \sigma_r^2(U_t))) \cdot \sigma_r^2(U_t) - 4\eta^2\lambda_1^3$$
$$\ge (1 + \eta\Delta/2 + 2\eta(\lambda_r - \Delta/2 - \sigma_1^2(J_t) - \sigma_r^2(U_t))) \cdot \sigma_r^2(U_t),$$

where we use $\sigma_r^2(U_t) \ge \Delta/4$ and $\eta \le \frac{\Delta^2}{32\lambda_1^3}$ in the second inequality. A combination of the above two inequalities gives that

$$\frac{\sigma_1^2(J_{t+1})}{\sigma_r^2(U_{t+1})} \le \frac{1 - \eta\Delta/2 + 2\eta(\lambda_r - \Delta/2 - \sigma_1^2(J_t) - \sigma_r^2(U_t))}{1 + \eta\Delta/2 + 2\eta(\lambda_r - \Delta/2 - \sigma_1^2(J_t) - \sigma_r^2(U_t))} \cdot \frac{\sigma_1^2(J_t)}{\sigma_r^2(U_t)}.$$

Since the function $g_7(s) = \frac{1 - \eta\Delta/2 + s}{1 + \eta\Delta/2 + s}$ is increasing on $[-1/2, 1/2]$, the condition $-1/2 \le 2\eta(\lambda_r - \Delta/2 - \sigma_1^2(\boldsymbol{J}_t) - \sigma_r^2(\boldsymbol{U}_t)) \le 1/2$ implies that

$$\frac{\sigma_1^2(\boldsymbol{J}_{t+1})}{\sigma_r^2(\boldsymbol{U}_{t+1})} \le \frac{3/2 - \eta\Delta/2}{3/2 + \eta\Delta/2} \cdot \frac{\sigma_1^2(\boldsymbol{J}_t)}{\sigma_r^2(\boldsymbol{U}_t)} \le (1 - \eta\Delta/3) \cdot \frac{\sigma_1^2(\boldsymbol{J}_t)}{\sigma_r^2(\boldsymbol{U}_t)}.$$

By deduction, we have

$$\sigma_1^2(\boldsymbol{J}_t) \le (1 - \eta\Delta/3)^t \cdot \sigma_r^2(\boldsymbol{U}_t)\frac{\sigma_1^2(\boldsymbol{J}_0)}{\sigma_r^2(\boldsymbol{U}_0)} \le (1 - \eta\Delta/3)^t \cdot \frac{8\lambda_1^2}{\Delta},$$

where the second inequality follows from $\sigma_r^2(\boldsymbol{U}_t) \le 2\lambda_1$, $\sigma_1^2(\boldsymbol{J}_0) \le \lambda_1$, and $\sigma_r^2(\boldsymbol{U}_0) \ge \Delta/4$. Therefore, for any $\epsilon > 0$, it takes at most $T_{\boldsymbol{J}}^\epsilon = \mathcal{O}(\frac{3}{\eta\Delta}\log\frac{8\lambda_1^2}{\epsilon\Delta})$ iterations to have $\sigma_1^2(\boldsymbol{J}_t) \le \epsilon$. ∎

### A.3 FINAL CONVERGENCE

For the convergence of $\boldsymbol{X}_t\boldsymbol{X}_t^\top$ to $\boldsymbol{\Sigma}_r$, It remains to show that $\boldsymbol{U}_t\boldsymbol{U}_t^\top$ converges to $\boldsymbol{\Lambda}_r$ fast, where $\boldsymbol{\Lambda}_r = \mathrm{diag}(\lambda_1, \ldots, \lambda_r)$. Equivalently, it suffices to show that $\sigma_1(\boldsymbol{P}_t)$ converges to zero linearly, where $\boldsymbol{P}_t = \boldsymbol{\Lambda}_t - \boldsymbol{U}_t\boldsymbol{U}_t^\top$. This is established in Lemma 20.

**Lemma 20** *Suppose $\eta \le \Delta^2/(36\lambda_1^3)$ and $\boldsymbol{X}_0 \in \mathcal{R}$. Then, for all $t \ge 0$, we have*

$$\sigma_1(\boldsymbol{P}_{t+1}) \le \frac{100\lambda_1^2}{\eta\Delta^2}(1 - \eta\Delta/4)^{t+1}.$$

*Hence, for any $\epsilon > 0$, it takes $T_{\boldsymbol{P}}^\epsilon = \mathcal{O}\left(\frac{4}{\eta\Delta}\log\frac{100\lambda_1^2}{\eta\Delta^2\epsilon}\right)$ iterations to reach $\sigma_1(\boldsymbol{P}_t) \le \epsilon$.*

**Proof** By Lemma 12, $\boldsymbol{X}_t \in \mathcal{R}$ for all $t \ge 0$. Using the notation of $\boldsymbol{P}_t$, (18) can be rewritten as

$$\boldsymbol{U}_{t+1} = \boldsymbol{U}_t + \eta\boldsymbol{P}_t\boldsymbol{U}_t - \eta\boldsymbol{U}_t\boldsymbol{J}_t^\top\boldsymbol{J}_t.$$

By direct calculation, we have

$$\boldsymbol{P}_{t+1} = (\boldsymbol{I}_r - \eta\boldsymbol{U}_t\boldsymbol{U}_t^\top)\boldsymbol{P}_t(\boldsymbol{I}_r - \eta\boldsymbol{U}_t\boldsymbol{U}_t^\top) - \eta^2(\boldsymbol{P}_t\boldsymbol{U}_t\boldsymbol{U}_t^\top\boldsymbol{P}_t + \boldsymbol{U}_t\boldsymbol{U}_t^\top\boldsymbol{P}_t\boldsymbol{U}_t\boldsymbol{U}_t^\top) + \boldsymbol{R}_t,$$

where

$$\boldsymbol{R}_t = \eta(\boldsymbol{I}_r + \eta\boldsymbol{P}_t)\boldsymbol{U}_t\boldsymbol{J}_t^\top\boldsymbol{J}_t\boldsymbol{U}_t^\top + \eta\boldsymbol{U}_t\boldsymbol{J}_t^\top\boldsymbol{J}_t\boldsymbol{U}_t^\top(\boldsymbol{I}_r + \eta\boldsymbol{P}_t) - \eta^2\boldsymbol{U}_t(\boldsymbol{J}_t^\top\boldsymbol{J}_t)^2\boldsymbol{U}_t^\top.$$

By the singular value inequality,

$$\sigma_1(\boldsymbol{P}_{t+1}) \le ((1 - \eta\Delta/4)^2 + 8\eta^2\lambda_1^2) \cdot \sigma_1(\boldsymbol{P}_t) + \sigma_1(\boldsymbol{R}_t)$$
$$\le (1 - \eta\Delta/4) \cdot \sigma_1(\boldsymbol{P}_t) + \sigma_1(\boldsymbol{R}_t),$$

where we use $\Delta/4 \le \sigma_r^2(\boldsymbol{U}_t) \le \sigma_1^2(\boldsymbol{U}_t) \le 2\lambda_1$ in the first inequality and $\eta \le \frac{\Delta}{36\lambda_1^2}$ in the second inequality. For the remainder term $\boldsymbol{R}_t$, by the singular value inequality and the condition $\eta \le \frac{\Delta^2}{36\lambda_1^3}$, we have

$$\sigma_1(\boldsymbol{R}_t) \le \sigma_1^2(\boldsymbol{J}_t) \le (1 - \eta\Delta/3)^t \cdot \frac{8\lambda_1^2}{\Delta},$$

where the last inequality follows from Lemma 19. Then by deduction, we have

$$\frac{\sigma_1(\boldsymbol{P}_{t+1})}{(1 - \eta\Delta/4)^{t+1}} \le \frac{\sigma_1(\boldsymbol{P}_t)}{(1 - \eta\Delta/4)^t} + \left(\frac{1 - \eta\Delta/3}{1 - \eta\Delta/4}\right)^t \frac{8\lambda_1^2}{(1 - \eta\Delta/4)\Delta}$$
$$\le \sigma_1(\boldsymbol{P}_0) + \sum_{i=1}^t \left(\frac{1 - \eta\Delta/3}{1 - \eta\Delta/4}\right)^i \frac{8\lambda_1^2}{(1 - \eta\Delta/4)\Delta}$$
$$\le \sigma_1(\boldsymbol{P}_0) + \frac{96\lambda_1^2}{\eta\Delta^2} \le \frac{100\lambda_1^2}{\eta\Delta^2},$$

where the last inequality follows from $\sigma_1(\boldsymbol{P}_0) \le 2\lambda_1$. Therefore, it takes $T_{\boldsymbol{P}}^\epsilon = \mathcal{O}(\frac{4}{\eta\Delta}\log\frac{100\lambda_1^2}{\eta\Delta^2\epsilon})$ iterations to achieve $\sigma_1(\boldsymbol{P}_t) \le \epsilon$. ∎

### A.4 PROOF OF THEOREM 2

By combining Lemma 19 and Lemma 20, we can prove Theorem 2.

**Proof** Observe that

$$\|\boldsymbol{\Sigma}_r - \boldsymbol{X}_t \boldsymbol{X}_t^\top\|_F \leq \|\boldsymbol{P}_t\|_F + 2\|\boldsymbol{J}_t \boldsymbol{X}_t^\top\|_F \leq r\sigma_1(\boldsymbol{P}_t) + 2r\sqrt{2\lambda_1}\sigma_1(\boldsymbol{J}_t), \quad \forall \boldsymbol{X}_t \in \mathcal{R},$$

where we use the fact that $\|\boldsymbol{A}\|_F \leq r\sigma_1(\boldsymbol{A})$ for any rank-$r$ matrix $\boldsymbol{A}$. Let

$$T^\epsilon = \max\left\{T_{\boldsymbol{J}}^{\epsilon^2/(32r^2\lambda_1)}, T_{\boldsymbol{P}}^{\epsilon/(2r)}\right\}.$$

Then, $\|\boldsymbol{\Sigma}_r - \boldsymbol{X}_t \boldsymbol{X}_t^\top\|_F \leq \epsilon$ for all $t \geq T^\epsilon$. Theorem 2 follows from $T^\epsilon = \mathcal{O}(\frac{6}{\eta\Delta}\log\frac{200r\lambda_1^2}{\eta\Delta^2\epsilon})$. ∎

## B  ANALYSIS OF LARGE INITIALIZATION

In this section, we will prove Theorem 6 as well as the results in Section 5.1. Before delving further, we first write down the update rules of $\boldsymbol{u}_{k,t}$ and $\boldsymbol{K}_{k,t}$. Recall that $\boldsymbol{u}_{k,t}$ and $\boldsymbol{K}_{k,t}$ are the $k$-th and $(k+1)$-to-$d$-th rows of $\boldsymbol{X}_t$. The update rules are given by

$$\boldsymbol{u}_{k,t+1} = \boldsymbol{u}_{k,t} + \eta\lambda_k \boldsymbol{u}_{k,t} - \eta\boldsymbol{u}_{k,t}\boldsymbol{X}_t^\top\boldsymbol{X}_t, \tag{22}$$

$$\boldsymbol{K}_{k,t+1} = \boldsymbol{K}_{k,t} + \eta\boldsymbol{\Gamma}_k \boldsymbol{K}_{k,t} - \eta\boldsymbol{K}_{k,t}\boldsymbol{X}_t^\top\boldsymbol{X}_t, \tag{23}$$

where $\boldsymbol{\Gamma}_k = \mathrm{diag}(\lambda_{k+1}, \ldots, \lambda_d)$. We also remind readers that $\boldsymbol{u}_{k,t} \in \mathbb{R}^{1\times r}$ is a row vector. Moreover, we let $\boldsymbol{\Pi}_{\boldsymbol{u}_{k,t}}$ denote the projection matrix associated with $\boldsymbol{u}_{k,t}$, that is,

$$\boldsymbol{\Pi}_{\boldsymbol{u}_{k,t}} = \boldsymbol{u}_{k,t}^\top(\boldsymbol{u}_{k,t}\boldsymbol{u}_{k,t}^\top)^{-1}\boldsymbol{u}_{k,t} \in \mathbb{R}^{r\times r}.$$

Also, we let $\boldsymbol{G}_{k,t}$ denote the first $k$ rows of $\boldsymbol{X}_t$.

### B.1  PROOFS FOR SECTION 5.1: RANK-TWO MATRIX APPROXIMATION

In this section, we collect proofs related to the rank-two matrix approximation.

#### B.1.1  PROOF OF LEMMA 8

**Proof** Note that $t_{\mathrm{init},1} \leq T_1 + T_{\boldsymbol{K}}$, where

$$T_1 = \min\{t \geq 0 \mid \sigma_1^2(\boldsymbol{X}_t) \leq 2\lambda_1\}$$

is the first time when $\sigma_1^2(\boldsymbol{X}_t)$ is smaller than $2\lambda_1$, and

$$T_{\boldsymbol{K}} = \min\{t \geq 0 \mid \sigma_1^2(\boldsymbol{K}_{k,t+T_1}) \leq \lambda_k - \frac{3\Delta}{4}, \forall k \leq r\}.$$

To prove the lemma, it suffices to analyze $T_1$ and $T_{\boldsymbol{K}}$ separately.

First, we analyze $T_1$ as follows.

- If $\sigma_1^2(\boldsymbol{X}_0) \leq 2\lambda_1$, then $T_1 = 0$.
- If $2\lambda_1 < \sigma_1^2(\boldsymbol{X}_0) < 1/(3\eta)$, then by Lemma 13, $\sigma_1^2(\boldsymbol{X}_t) \leq 1/(3\eta)$ for all $t$. Furthermore, it follows from Lemma 16 that

$$\sigma_1(\boldsymbol{X}_{t+1}) \leq (1 + \eta\lambda_1 - \eta\sigma_1^2(\boldsymbol{X}_t)) \cdot \sigma_1(\boldsymbol{X}_t)$$
$$\leq (1 - \eta\lambda_1) \cdot \sigma_1(\boldsymbol{X}_t), \quad \forall t < T_1,$$

where the second inequality uses $\sigma_1^2(\boldsymbol{X}_t) > 2\lambda_1$ for all $t < T_1$. It implies that

$$\sigma_1(\boldsymbol{X}_t) \leq (1 - \eta\lambda_1)^t \cdot \sigma_1(\boldsymbol{X}_0)$$

for all $t \leq T_1$ and

$$T_1 = \mathcal{O}\left(\frac{1}{\eta\lambda_1}\log\frac{\sigma_1(\boldsymbol{X}_0)}{\sqrt{2\lambda_1}}\right).$$

By Lemma 13, we have $\sigma_1^2(\boldsymbol{X}_t) \leq 2\lambda_1$ for all $t \geq T_1$.

Next, we analyze $T_{\boldsymbol{K}}$ and the following quantities

$$T_{\boldsymbol{K}_k} = \min\{t \geq 0 \mid \sigma_1^2(\boldsymbol{K}_{k,t+T_1}) \leq \lambda_k - \frac{3\Delta}{4}\}.$$

Recall that $\boldsymbol{K}_{k,t}$ is the $(k+1)$-to-$d$-th rows of $\boldsymbol{X}_t$. Then by (23), we have

$$\boldsymbol{K}_{k,t+1} = \boldsymbol{K}_{k,t} + \eta\boldsymbol{\Gamma}_k\boldsymbol{K}_{k,t} - \eta\boldsymbol{K}_{k,t}\boldsymbol{X}_t^\top\boldsymbol{X}_t$$
$$= \underbrace{\frac{1}{2}\boldsymbol{K}_{k,t} - \eta\boldsymbol{K}_{k,t}\boldsymbol{K}_{k,t}^\top\boldsymbol{K}_{k,t}}_{\boldsymbol{B}} + \underbrace{(\frac{1}{4}\boldsymbol{I}_{d-k} + \eta\boldsymbol{\Gamma}_k)\boldsymbol{K}_{k,t}}_{\boldsymbol{C}} + \underbrace{\boldsymbol{K}_{k,t}(\frac{1}{4}\boldsymbol{I}_k - \eta\boldsymbol{G}_{k,t}^\top\boldsymbol{G}_{k,t})}_{\boldsymbol{D}},$$

where $\boldsymbol{\Gamma}_k = \mathrm{diag}(\lambda_{k+1}, \ldots, \lambda_d)$ and $\boldsymbol{G}_{k,t} \in \mathbb{R}^{k \times r}$ is the first $k$ rows of $\boldsymbol{X}_t$. By the singular value inequality, we obtain

$$\sigma_1(\boldsymbol{K}_{k,t+1}) \leq \sigma_1(\boldsymbol{B}) + \sigma_1(\boldsymbol{C}) + \sigma_1(\boldsymbol{D}).$$

For the first term $\boldsymbol{B}$, similar to Lemma 17, we can show that

$$\sigma_1(\boldsymbol{B}) = \sigma_1(\boldsymbol{K}_{k,t})/2 - \eta\sigma_1^3(\boldsymbol{K}_{k,t}), \quad \forall t \geq T_1.$$

For the second term $\boldsymbol{C}$, by the singular value inequality,

$$\sigma_1(\boldsymbol{C}) \leq (\frac{1}{4} + \eta\lambda_{k+1}) \cdot \sigma_1(\boldsymbol{K}_{k,t}).$$

For the third term $\boldsymbol{D}$, since $\boldsymbol{G}_{k,t}^\top\boldsymbol{G}_{k,t}$ is PSD and $\eta\sigma_1^2(\boldsymbol{G}_{k,t}) \leq \frac{1}{4}$ for all $t \geq T_1$, we have

$$\sigma_1(\boldsymbol{D}) \leq \sigma_1(\boldsymbol{K}_{k,t})/4, \quad \forall t \geq T_1.$$

Combining,

$$\sigma_1(\boldsymbol{K}_{k,t+1}) \leq (1 + \eta\lambda_{k+1} - \eta\sigma_1^2(\boldsymbol{K}_{k,t})) \cdot \sigma_1(\boldsymbol{K}_{k,t}), \quad \forall t \geq T_1, \quad \forall k \leq r. \quad (24)$$

Since $\lambda_{k+1} \leq \lambda_k - \Delta$ for $k \leq r$, (24) implies that

$$\sigma_1(\boldsymbol{K}_{k,t+T_1+1}) \leq (1 - \eta\Delta/4) \cdot \sigma_1(\boldsymbol{K}_{k,t+T_1}), \quad \forall t < T_{\boldsymbol{K}_k}, \quad \forall k \leq r.$$

Hence, $\sigma_1(\boldsymbol{K}_{k,t+T_1}) \leq (1 - \eta\Delta/4)^t \cdot \sigma_1(\boldsymbol{K}_{k,T_1})$ for all $t \leq T_{\boldsymbol{K}_k}$. In particular,

$$T_{\boldsymbol{K}_k} = \mathcal{O}\left(\frac{2}{\eta\Delta} \log \frac{\sigma_1^2(\boldsymbol{K}_{k,T_1})}{\lambda_k - \frac{3\Delta}{4}}\right) \text{ and } T_{\boldsymbol{K}} = \mathcal{O}\left(\frac{2}{\eta\Delta} \log \frac{8\lambda_1}{\Delta}\right),$$

where we use $\sigma_1^2(\boldsymbol{K}_{k,T_1}) \leq 2\lambda_1$ and $\lambda_k - \frac{3\Delta}{4} \geq \frac{\Delta}{4}$.

Finally, similar to Lemma 13 and 14, for any $a \geq \lambda_{k+1}$, if $\sigma_1^2(\boldsymbol{K}_{k,t+T_1}) \leq a$, then $\sigma_1^2(\boldsymbol{K}_{k,t'+T_1}) \leq a$ for all $t' \geq t$. This implies that $\sigma_1^2(\boldsymbol{K}_{k,t+T_1}) \leq \lambda_k - \frac{3\Delta}{4}$ for all $t \geq T_{\boldsymbol{K}}$ for $k \leq r$. ∎

### B.1.2 PROOF OF LEMMA 9

**Proof** This lemma is a special case of Lemma 21, where we take $k = 1$ and $t_{\mathrm{init}} = t_{\mathrm{init},1}$. Notice that $\boldsymbol{G}_{0,t} = \boldsymbol{0}$ and $\boldsymbol{X}_t \in \mathcal{S}$ for all $t \geq t_{\mathrm{init},1}$ by Lemma 8. Thus, the conditions in Lemma 21 trivially hold. Then Lemma 9 immediately follows from Lemma 21. ∎

### B.1.3 PROOF OF LEMMA 10

**Proof** The lemma is a special case of Lemma 22 and Lemma 23. In Lemma 22, we take $k = 1$ and $t_{\mathrm{init}} = t_{\mathrm{init},1} + T_{\boldsymbol{u}_1}$. In Lemma 23, we take $k = 1$ and $t_{\mathrm{init}} = t_{\mathrm{init},1} + T_{\boldsymbol{u}_1} + t^*$. ∎

### B.1.4 PROOF OF LEMMA 11

**Proof** This lemma is a special case of Lemma 21, where we take $k = 2$ and $t_{\mathrm{init}} = t_1 + t_1^*$. ∎

### B.2 PROOF OF THEOREM 6

**Proof** The first property follows from Lemma 8.

To prove the remaining properties in this theorem, we will use an inductive argument. Our induction hypotheses are listed below:

H$(k,1)$ $\sigma_1(\boldsymbol{u}_{k,t}\boldsymbol{G}_{k-1,t}^\top) \leq \sqrt{\frac{\Delta}{8}}\min\{\sigma_1(\boldsymbol{u}_{k,t_{\text{init},k}}), \sqrt{\frac{\Delta}{2}}\} \cdot (1 - \eta\Delta/6)^{t-t_{\text{init},k}}$ for all $t \geq t_{\text{init},k}$.

H$(k,2)$ $T_{\boldsymbol{u}_k} = \mathcal{O}\left(\frac{4}{\eta\Delta}\log\frac{\Delta}{2\sigma_1^2(\boldsymbol{u}_{k,t_{\text{init},k}})}\right)$ and $\sigma_1^2(\boldsymbol{u}_{k,t}) \geq \frac{\Delta}{2}$ for all $t \geq t_{\text{init},k} + T_{\boldsymbol{u}_k}$.

H$(k,3)$ $\sigma_1(\boldsymbol{u}_{k,t}\boldsymbol{K}_{k,t}^\top) \leq (1 - \eta\Delta/6)^{t-t_k}$ for all $t \geq t_k$.

Note that H(1,1) trivially holds because $\boldsymbol{G}_{0,t} = \boldsymbol{0}$. Then we prove H$(k,1)$, H$(k,2)$, H$(k,3)$, H$(k+1,1)$ successively until H$(r,3)$.

- $\{\text{H}(j,\cdot)\}_{j<k} + \text{H}(k,1) \rightarrow \text{H}(k,2)$

  This follows from Lemma 21, where we take $t_{\text{init}} = t_{\text{init},k}$.

- $\{\text{H}(j,\cdot)\}_{j<k} + \text{H}(k,1) + \text{H}(k,2) \rightarrow \text{H}(k,3)$

  This follows from Lemma 22, where we take $t_{\text{init}} = t_{\text{init},k} + T_{\boldsymbol{u}_k}$.

- $\{\text{H}(j,\cdot)\}_{j\leq k} \rightarrow \text{H}(k+1,1)$

  By $\{\text{H}(j,3)\}_{j\leq k}$,

$$\sigma_1(\boldsymbol{u}_{k+1,t}\boldsymbol{G}_{k,t}^\top) \leq \sum_{j\leq k}\sigma_1(\boldsymbol{u}_{j,t}\boldsymbol{K}_{j,t}^\top) \leq r(1 - \eta\Delta/6)^{t-t_k},$$

  for all $t \geq t_k$. By definition of $t_k^*$, we have

$$r(1 - \eta\Delta/6)^{t_k^*} \leq \sqrt{\frac{\Delta}{8}}\min\{\sigma_1(\boldsymbol{u}_{k,t_k+t_k^*}), \sqrt{\frac{\Delta}{2}}\}.$$

  Then H$(k+1,1)$ follows from the definition $t_{\text{init},k+1} = t_k + t_k^*$.

By induction, H$(k,\cdot)$ holds for all $k \leq r$.

For all $t \geq t_k$, (16) follows from Lemma 23, where $t_{\text{init}}$ is taken as $t_k$.

For all $t \geq t_{\text{init},r} + T_{\boldsymbol{u}_r}$, we have $\sigma_1^2(\boldsymbol{u}_{k,t}) \geq \frac{\Delta}{2}$ for all $k \leq r$. Simultaneously,

$$\sum_{j\leq r}\sigma_1(\boldsymbol{u}_{j,t}\boldsymbol{K}_{j,t}^\top) \leq r(1 - \eta\Delta/6)^{t-(t_{\text{init},r}+T_{\boldsymbol{u}_r}+t^*)}$$

holds for all $t \geq t_{\text{init},r} + T_{\boldsymbol{u}_r} + t^*$. Let $\boldsymbol{U}_t$ be the first $r$ rows of $\boldsymbol{X}_t$. Viewing $\boldsymbol{U}_t\boldsymbol{U}_t^\top$ as the sum of diagonal elements and off-diagonal elements, we find that

$$\sigma_r^2(\boldsymbol{U}_t) \geq \Delta/2 - r(1 - \eta\Delta/6)^{t-(t_{\text{init},r}+T_{\boldsymbol{u}_r}+t^*)}$$

for all $t \geq t_{\text{init},r} + T_{\boldsymbol{u}_r} + t^*$. Hence, $\sigma_r^2(\boldsymbol{U}_t) \geq \Delta/4$ for all $t \geq t_{\text{init},r} + T_{\boldsymbol{u}_r} + t^* + t^\sharp$, where

$$t^\sharp = \frac{\log(\Delta/(4r))}{\log(1 - \eta\Delta/6)}.$$

This implies that $\boldsymbol{X}_t \in \mathcal{R}$ for $t \geq t_{\text{init},r} + T_{\boldsymbol{u}_r} + t^* + t^\sharp$.

The sixth property is merely an application of Theorem 2.

The seventh property immediately follows from the previous six properties. ∎

### B.3 TECHNICAL LEMMAS

This section collects technical lemmas that are used in previous sections. Let us recall that $\boldsymbol{u}_{k,t}$ and $\boldsymbol{K}_{k,t}$ are the $k$-th and the $(k+1)$-to-$d$-th rows of $\boldsymbol{X}_t$ respectively. The projection matrix associated with $\boldsymbol{u}_{k,t}$ is denoted by

$$\boldsymbol{\Pi}_{\boldsymbol{u}_{k,t}} = \boldsymbol{u}_{k,t}^\top (\boldsymbol{u}_{k,t}\boldsymbol{u}_{k,t}^\top)^{-1}\boldsymbol{u}_{k,t}.$$

The first $k$ rows of $\boldsymbol{X}_t$ are denoted by $\boldsymbol{G}_{k,t}$, and $\boldsymbol{G}_{0,t} = \boldsymbol{0}$ by definition.

#### B.3.1 DYNAMICS

This subsection contains lemmas describing the dynamics of the GD sequence.

Lemma 21 shows that when $\sigma_1(\boldsymbol{u}_{k,t}\boldsymbol{G}_{k-1,t}^\top)$ is sufficiently small, the signal term $\sigma_1^2(\boldsymbol{u}_{k,t+1})$ can rise above $\Delta/2$ quickly. Moreover, as shown in Lemma 21, the term $\sigma_1^2(\boldsymbol{u}_{k,t+1})$ will remain larger than $\Delta/2$.

**Lemma 21** *Suppose $\eta \le \frac{1}{12\lambda_1}$, $\boldsymbol{X}_t \in \mathcal{S}$, and for some $t_{\text{init}} \ge 0$ and $k \le r$, the condition*

$$\sigma_1(\boldsymbol{u}_{k,t}\boldsymbol{G}_{k-1,t}^\top) \le \sqrt{\frac{\Delta}{8}} \min\{\sigma_1(\boldsymbol{u}_{k,t_{\text{init}}}), \sqrt{\frac{\Delta}{2}}\} \cdot (1 - \eta\Delta/6)^{t-t_{\text{init}}}$$

*holds for all $t \ge t_{\text{init}}$. Then $\sigma_1^2(\boldsymbol{u}_{k,t}) \ge \frac{\Delta}{2}$ for all $t \ge t_{\text{init}} + T_{\boldsymbol{u}_k}$, where*

$$T_{\boldsymbol{u}_k} = \mathcal{O}\left(\frac{4}{\eta\Delta}\log\frac{\Delta}{2\sigma_1^2(\boldsymbol{u}_{k,t_{\text{init}}})}\right).$$

*In addition, for all $t \ge t_{\text{init}}$, we have*

$$\sigma_1^2(\boldsymbol{u}_{k,t+1}) \ge (1 + 2\eta\lambda_k - \eta\Delta/4 - 2\eta\sigma_1^2(\boldsymbol{u}_{k,t}) - 2\eta\sigma_1^2(\boldsymbol{K}_{k,t}\boldsymbol{\Pi}_{\boldsymbol{u}_{k,t}}))) \cdot \sigma_1^2(\boldsymbol{u}_{k,t}), \tag{25}$$

*where $\boldsymbol{\Pi}_{\boldsymbol{u}_{k,t}} = \boldsymbol{u}_{k,t}^\top(\boldsymbol{u}_{k,t}\boldsymbol{u}_{k,t}^\top)^{-1}\boldsymbol{u}_{k,t}$ is the projection matrix associated with $\boldsymbol{u}_{k,t}$.*

**Proof** First, we show that $\sigma_1^2(\boldsymbol{u}_{k,t}) \ge \min\{\sigma_1^2(\boldsymbol{u}_{k,t_{\text{init}}}), \frac{\Delta}{2}\}$ for all $t \ge t_{\text{init}}$ by induction.

This is true when $t = t_{\text{init}}$. Now suppose $\sigma_1^2(\boldsymbol{u}_{k,t}) \ge \min\{\sigma_1^2(\boldsymbol{u}_{k,t_{\text{init}}}), \frac{\Delta}{2}\}$ for some $t \ge t_{\text{init}}$. By assumption, $\sigma_1^2(\boldsymbol{u}_{k,t}\boldsymbol{G}_{k-1,t}^\top) \le \frac{\Delta}{8}\min\{\sigma_1^2(\boldsymbol{u}_{k,t_{\text{init}}}), \frac{\Delta}{2}\} \le \frac{\Delta}{8}\sigma_1^2(\boldsymbol{u}_{k,t})$. Then by Lemma 24 and $\boldsymbol{X}_t \in \mathcal{S}$, we have

$$\sigma_1^2(\boldsymbol{u}_{k,t+1}) \ge (1 + 2\eta\lambda_k - 2\eta\sigma_1^2(\boldsymbol{u}_{k,t}) - 2\eta\sigma_1^2(\boldsymbol{K}_{k,t}\boldsymbol{\Pi}_{\boldsymbol{u}_{k,t}}))) \cdot \sigma_1^2(\boldsymbol{u}_{k,t}) - \frac{\eta\Delta}{4}\sigma_1^2(\boldsymbol{u}_{k,t}) \tag{26}$$

$$\ge (1 + 5\eta\Delta/4 - 2\eta\sigma_1^2(\boldsymbol{u}_{k,t})) \cdot \sigma_1^2(\boldsymbol{u}_{k,t}). \tag{27}$$

Then we consider two cases.

- If $\sigma_1^2(\boldsymbol{u}_{k,t}) \le \frac{5\Delta}{8}$, then $\sigma_1^2(\boldsymbol{u}_{k,t+1}) \ge \sigma_1^2(\boldsymbol{u}_{k,t}) \ge \min\{\sigma_1^2(\boldsymbol{u}_{k,t_{\text{init}}}), \frac{\Delta}{2}\}$.
- If $\sigma_1^2(\boldsymbol{u}_{k,t}) \ge \frac{5\Delta}{8}$, then

$$\sigma_1^2(\boldsymbol{u}_{k,t+1}) \ge (1 + \frac{5\eta\Delta}{4} - \frac{5\eta\Delta}{4}) \cdot \frac{5\Delta}{8} = \frac{5\Delta}{8}$$

$$\ge \min\{\sigma_1^2(\boldsymbol{u}_{k,t_{\text{init}}}), \frac{\Delta}{2}\},$$

  where the first inequality uses the fact that $g_8(s) = (1 + \frac{5\eta\Delta}{4} - 2\eta s)s$ is increasing on $(-\infty, 1/4\eta]$.

In both cases, we have $\sigma_1^2(\boldsymbol{u}_{k,t+1}) \ge \min\{\sigma_1^2(\boldsymbol{u}_{k,\text{init}}), \frac{\Delta}{2}\}$. The claim then follows by induction.

Furthermore, the above analysis shows that inequalities 26 and 27 hold for all $t \ge t_{\text{init}}$, which leads to the inequality 25.

Let

$$T_{\boldsymbol{u}_k} = \min\{t \geq 0 \mid \sigma_1^2(\boldsymbol{u}_{k,t+t_{\text{init}}}) \geq \frac{\Delta}{2}\}.$$

Then for $t < T_{\boldsymbol{u}_k}$, we have $\sigma_1^2(\boldsymbol{u}_{k,t+t_{\text{init}}}) < \frac{\Delta}{2}$ and by inequality 27,

$$\sigma_1^2(\boldsymbol{u}_{k,t+1+t_{\text{init}}}) \geq (1 + \eta\Delta/4) \cdot \sigma_1^2(\boldsymbol{u}_{k,t+t_{\text{init}}}).$$

Hence, for all $t \leq T_{\boldsymbol{u}_k}$, we have

$$\sigma_1^2(\boldsymbol{u}_{k,t+t_{\text{init}}}) \geq (1 + \eta\Delta/4)^t \cdot \sigma_1^2(\boldsymbol{u}_{k,t_{\text{init}}}),$$

and

$$T_{\boldsymbol{u}_k} = \mathcal{O}\left(\frac{4}{\eta\Delta} \log \frac{\Delta}{2\sigma_1^2(\boldsymbol{u}_{k,t_{\text{init}}})}\right).$$

Finally, by inequality 27, we have for any $a \leq \frac{5\Delta}{8}$, if $\sigma_1^2(\boldsymbol{u}_{k,t}) \geq a$, then $\sigma_1^2(\boldsymbol{u}_{k,t+1}) \geq a$. Thus, by induction, $\sigma_1^2(\boldsymbol{u}_{k,t}) \geq \frac{\Delta}{2}$ for all $t \geq t_{\text{init}} + T_{\boldsymbol{u}_k}$. ∎

Lemma 22 shows that when the noise terms $\sigma_1(\boldsymbol{u}_{j,t}\boldsymbol{K}_{j,t}^\top)$ converge linearly to zero for all $j < k$ and the $k$-th signal term $\sigma_1^2(\boldsymbol{u}_{k,t}) \geq \frac{\Delta}{2}$, the noise term $\sigma_1(\boldsymbol{u}_{k,t}\boldsymbol{K}_{k,t}^\top)$ will also converge linearly to zero. The key component is to analyze the SNR $\frac{\sigma_1^2(\boldsymbol{u}_{k,t})}{\sigma_1(\boldsymbol{u}_{k,t}\boldsymbol{K}_{k,t}^\top)}$.

**Lemma 22** *Suppose $\eta \leq \frac{\Delta}{100\lambda_1^2}$, $\boldsymbol{X}_t \in \mathcal{S}$, and for some $t_{\text{init}} \geq 0$ and $k \leq r$, the conditions*

$$\sigma_1(\boldsymbol{u}_{j,t}\boldsymbol{K}_{j,t}^\top) \leq (1 - \eta\Delta/6)^{t-t_{\text{init}}}, \quad \forall j < k, \tag{28}$$

$$\sigma_1(\boldsymbol{u}_{k,t}\boldsymbol{G}_{k-1,t}^\top) \leq \frac{\Delta}{4}(1 - \eta\Delta/6)^{t-t_{\text{init}}}, \tag{29}$$

$$\sigma_1^2(\boldsymbol{u}_{k,t}) \geq \frac{\Delta}{2} \tag{30}$$

*hold for all $t \geq t_{\text{init}}$. Then we have*

$$\sigma_1(\boldsymbol{u}_{k,t}\boldsymbol{K}_{k,t}^\top) \leq (1 - \eta\Delta/6)^{t-t_{\text{init}}-t^*}$$

*for all $t \geq t_{\text{init}} + t^*$, where*

$$t^* = \log\left(\frac{\Delta^2}{8\lambda_1^3 + 144r^2\lambda_1}\right) / \log(1 - \eta\Delta/6).$$

**Proof** By condition 29, we can apply Lemma 21 to obtain

$$\sigma_1^2(\boldsymbol{u}_{k,t+1}) \geq (1 + 2\eta\lambda_k - \eta\Delta/4 - 2\eta\sigma_1^2(\boldsymbol{u}_{k,t}) - 2\eta\sigma_1^2(\boldsymbol{K}_{k,t}\boldsymbol{\Pi}_{\boldsymbol{u}_{k,t}}))) \cdot \sigma_1^2(\boldsymbol{u}_{k,t})$$

for all $t \geq t_{\text{init}}$. By Lemma 25, we have

$$\begin{aligned}
&\sigma_1(\boldsymbol{u}_{k,t+1}\boldsymbol{K}_{k,t+1}^\top) \\
&\leq (1 + \eta\lambda_k + \eta\lambda_{k+1} - 2\eta\sigma_1^2(\boldsymbol{u}_{k,t}) - 2\eta\sigma_1^2(\boldsymbol{K}_{k,t}\boldsymbol{\Pi}_{\boldsymbol{u}_{k,t}}) + 25\eta^2\lambda_1^2) \cdot \sigma_1(\boldsymbol{u}_{k,t}\boldsymbol{K}_{k,t}^\top) \\
&\quad + 3\eta\sigma_1(\boldsymbol{u}_{k,t}\boldsymbol{G}_{k-1,t}^\top)\sigma_1(\boldsymbol{K}_{k,t}\boldsymbol{G}_{k-1,t}^\top)
\end{aligned}$$

for all $t \geq t_{\text{init}}$. Divide both sides of the inequality by $\sigma_1^2(\boldsymbol{u}_{k,t+1})$. By Lemma 26 and $\sigma_1^2(\boldsymbol{u}_{k,t+1}) \geq \frac{\Delta}{2}$, we have

$$\frac{\sigma_1(\boldsymbol{u}_{k,t+1}\boldsymbol{K}_{k,t+1}^\top)}{\sigma_1^2(\boldsymbol{u}_{k,t+1})} \leq (1 - \eta\Delta/6)\frac{\sigma_1(\boldsymbol{u}_{k,t}\boldsymbol{K}_{k,t}^\top)}{\sigma_1^2(\boldsymbol{u}_{k,t})} + \frac{6\eta}{\Delta}\sigma_1(\boldsymbol{u}_{k,t}\boldsymbol{G}_{k-1,t}^\top)\sigma_1(\boldsymbol{K}_{k,t}\boldsymbol{G}_{k-1,t}^\top) \tag{31}$$

for all $t \geq t_{\text{init}}$. Observe that by condition 28 and definitions of $\boldsymbol{u}_{k,t}$, $\boldsymbol{K}_{k,t}$, and $\boldsymbol{G}_{k-1,t}$, we have

$$\max\{\sigma_1(\boldsymbol{u}_{k,t}\boldsymbol{G}_{k-1,t}^\top), \sigma_1(\boldsymbol{K}_{k,t}\boldsymbol{G}_{k-1,t}^\top)\} \leq \sum_{j<k}\sigma_1(\boldsymbol{u}_{k,t}\boldsymbol{K}_{k,t}^\top) \leq r(1 - \eta\Delta/6)^{t-t_{\text{init}}} \tag{32}$$

for all $t \geq t_{\mathrm{init}}$. Combining (31) and (32),

$$\frac{\sigma_1(\boldsymbol{u}_{k,t+1}\boldsymbol{K}_{k,t+1}^\top)}{\sigma_1^2(\boldsymbol{u}_{k,t+1})} \leq (1 - \eta\Delta/6)\frac{\sigma_1(\boldsymbol{u}_{k,t}\boldsymbol{K}_{k,t}^\top)}{\sigma_1^2(\boldsymbol{u}_{k,t})} + \frac{6\eta r^2}{\Delta}(1 - \eta\Delta/6)^{2(t-t_{\mathrm{init}})}$$

for all $t \geq t_{\mathrm{init}}$. Therefore, for all $t \geq t_{\mathrm{init}}$,

$$Q_{t+1} \leq (1 - \eta\Delta/6) \cdot Q_t,$$

where the quantity $Q_t$ is given by

$$Q_t = \frac{\sigma_1(\boldsymbol{u}_{k,t}\boldsymbol{K}_{k,t}^\top)}{\sigma_1^2(\boldsymbol{u}_{k,t})} + \frac{36r^2}{\Delta^2}(1 - \eta\Delta/6)^{2(t-t_{\mathrm{init}})-1}.$$

By induction, we have

$$\frac{\sigma_1(\boldsymbol{u}_{k,t}\boldsymbol{K}_{k,t}^\top)}{\sigma_1^2(\boldsymbol{u}_{k,t})} \leq (1 - \eta\Delta/6)^{t-t_{\mathrm{init}}}\left(\frac{\sigma_1(\boldsymbol{u}_{k,t_{\mathrm{init}}}\boldsymbol{K}_{k,t_{\mathrm{init}}}^\top)}{\sigma_1^2(\boldsymbol{u}_{k,t_{\mathrm{init}}})} + \frac{36r^2}{\Delta^2}(1 - \eta\Delta/6)^{-1}\right).$$

This implies that

$$\sigma_1(\boldsymbol{u}_{k,t}\boldsymbol{K}_{k,t}^\top) \leq \frac{8\lambda_1^3 + 144r^2\lambda_1}{\Delta^2} \cdot (1 - \eta\Delta/6)^{t-t_{\mathrm{init}}},$$

where we use $1 - \eta\Delta/6 \geq 1/2$, $\sigma_1^2(\boldsymbol{X}_t) \leq 2\lambda_1$, and $\sigma_1^2(\boldsymbol{u}_{k,t}) \geq \frac{\Delta}{2}$ for all $t \geq t_{\mathrm{init}}$. By definition of $t^*$, we have $(1 - \eta\Delta/6)^{t^*} \leq \frac{\Delta^2}{8\lambda_1^3 + 144r^2\lambda_1}$. Hence, for all $t \geq t_{\mathrm{init}} + t^*$, we have

$$\sigma_1(\boldsymbol{u}_{k,t}\boldsymbol{K}_{k,t}^\top) \leq (1 - \eta\Delta/6)^{t-t_{\mathrm{init}}-t^*},$$

which concludes the proof. ∎

Let $p_{k,t} = \lambda_k - \sigma_1^2(\boldsymbol{u}_{k,t})$ be the error term associated with the $k$-th signal. Lemma 23 shows that when the noise terms $\sigma_1(\boldsymbol{u}_{j,t}\boldsymbol{K}_{j,t}^\top)$ converge linearly to zero for all $j \leq k$ and the $k$-th signal term $\sigma_1^2(\boldsymbol{u}_{k,t}) \geq \frac{\Delta}{2}$, this signal term will converge fast to $\lambda_k$. Specifically, the error term $|p_{k,t}|$ will converge to zero at a linear rate. The analysis is similar to Lemma 20.

**Lemma 23** *Suppose $\eta \leq \frac{\Delta}{100\lambda_1^2}$, $\boldsymbol{X}_t \in \mathcal{S}$, and for some $t_{\mathrm{init}} \geq 0$ and $k \leq r$, the conditions*

$$\sigma_1(\boldsymbol{u}_{j,t}\boldsymbol{K}_{j,t}^\top) \leq (1 - \eta\Delta/6)^{t-t_{\mathrm{init}}}, \quad \forall j \leq k, \tag{33}$$

$$\sigma_1^2(\boldsymbol{u}_{k,t}) \geq \frac{\Delta}{2} \tag{34}$$

*hold for all $t \geq t_{\mathrm{init}}$. Then for all $t \geq t_{\mathrm{init}}$, we have*

$$|p_{k,t}| \leq (2\lambda_1 + \frac{24r}{\eta\Delta}) \cdot (1 - \eta\Delta/8)^{t-t_{\mathrm{init}}},$$

*where $p_{k,t} = \lambda_k - \sigma_1^2(\boldsymbol{u}_{k,t})$.*

**Proof** Using the notation of $p_{k,t}$, (22) can be rewritten as

$$\boldsymbol{u}_{k,t+1} = \boldsymbol{u}_{k,t} + \eta p_{k,t}\boldsymbol{u}_{k,t} - \eta\boldsymbol{u}_{k,t}\boldsymbol{W}_t.$$

where

$$\boldsymbol{W}_t = \boldsymbol{G}_{k-1,t}^\top\boldsymbol{G}_{k-1,t} + \boldsymbol{K}_{k,t}^\top\boldsymbol{K}_{k,t}.$$

By direction calculation, we have

$$p_{k,t+1} = p_{k,t} \cdot ((1 - \eta\sigma_1^2(\boldsymbol{u}_{k,t}))^2 + \eta^2\lambda_k\sigma_1^2(\boldsymbol{u}_{k,t})) + \mathrm{res}_t$$

where

$$\mathrm{res}_t = 2\eta(1 + \eta p_{k,t})\boldsymbol{u}_{k,t}\boldsymbol{W}_t\boldsymbol{u}_{k,t}^\top - \eta^2\boldsymbol{u}_{k,t}\boldsymbol{W}_t^2\boldsymbol{u}_{k,t}^\top.$$

By the singular value inequality, for all $t \geq t_{\text{init}}$, we have

$$
\begin{aligned}
|p_{k,t+1}| &\leq |p_{k,t}| \cdot ((1 - \eta\sigma_1^2(\boldsymbol{u}_{k,t}))^2 + \eta^2\lambda_k\sigma_1^2(\boldsymbol{u}_{k,t})) + |\text{res}_t| \\
&\leq |p_{k,t}| \cdot ((1 - \eta\Delta/2)^2 + 2\eta^2\lambda_1^2) + |\text{res}_t| \\
&\leq |p_{k,t}| \cdot (1 - \eta\Delta/2) + |\text{res}_t|,
\end{aligned}
\tag{35}
$$

where the second inequality uses $\Delta/2 \leq \sigma_1^2(\boldsymbol{u}_{k,t}) \leq 2\lambda_1$ and the third inequality use $\eta \leq \frac{\Delta}{100\lambda_1^2}$. Using $\eta \leq \frac{\Delta}{100\lambda_1^2}$ and $\sigma_1^2(\boldsymbol{X}_t) \leq 2\lambda_1$, we have

$$
|\text{res}_t| \leq \sum_{j \leq k} \sigma_1(\boldsymbol{u}_{k,t}\boldsymbol{K}_{k,t}^\top) \leq r(1 - \eta\Delta/6)^{t-t_{\text{init}}}
$$

for all $t \geq t_{\text{init}}$. Substituting this into (35), we obtain

$$
\begin{aligned}
|p_{k,t+1}| &\leq |p_{k,t}| \cdot (1 - \eta\Delta/2) + r(1 - \eta\Delta/6)^{t-t_{\text{init}}} \\
&\leq |p_{k,t}| \cdot (1 - \eta\Delta/8) + r(1 - \eta\Delta/6)^{t-t_{\text{init}}}.
\end{aligned}
$$

This implies that for all $t \geq t_{\text{init}}$,

$$
Q_{t+1} \leq Q_t + \frac{r}{1 - \eta\Delta/8}\left(\frac{1 - \eta\Delta/6}{1 - \eta\Delta/8}\right)^{t-t_{\text{init}}},
$$

where

$$
Q_t = \frac{|p_{k,t}|}{(1 - \eta\Delta/8)^{t-t_{\text{init}}}}.
$$

By induction, for all $t \geq t_{\text{init}}$, we have

$$
\begin{aligned}
Q_t &\leq Q_{t_{\text{init}}} + \frac{r}{1 - \eta\Delta/8}\sum_{i=0}^{t-1-t_{\text{init}}}\left(\frac{1 - \eta\Delta/6}{1 - \eta\Delta/8}\right)^i \\
&\leq |p_{k,t_{\text{init}}}| + \frac{24r}{\eta\Delta} \\
&\leq 2\lambda_1 + \frac{24r}{\eta\Delta}.
\end{aligned}
$$

Hence, for all $t \geq t_{\text{init}}$, we have

$$
|p_{k,t}| \leq (2\lambda_1 + \frac{24r}{\eta\Delta}) \cdot (1 - \eta\Delta/8)^{t-t_{\text{init}}},
$$

which concludes the proof. ∎

### B.3.2 Technical calculations

The following lemmas provide calculations related to an SNR argument, where the SNR refers to the ratio $\frac{\sigma_1^2(\boldsymbol{u}_{k,t})}{\sigma_1(\boldsymbol{u}_{k,t}\boldsymbol{K}_{k,t}^\top)}$. Recall that $\boldsymbol{u}_{k,t}$ is the $k$-th row of $\boldsymbol{X}_t$ and $\boldsymbol{K}_{k,t}$ represents the $(k+1)$-to-$d$-th rows of $\boldsymbol{X}_t$. Moreover, we recall that

$$
\boldsymbol{\Pi}_{\boldsymbol{u}_{k,t}} = \boldsymbol{u}_{k,t}^\top(\boldsymbol{u}_{k,t}\boldsymbol{u}_{k,t}^\top)^{-1}\boldsymbol{u}_{k,t}
$$

is the projection matrix associated with $\boldsymbol{u}_{k,t}$. $\boldsymbol{G}_{k,t}$ collects the first $k$ rows of $\boldsymbol{X}_t$.

Lemma 24 provides a lower bound on $\sigma_1^2(\boldsymbol{u}_{k,t+1})$ in terms of the preceding iteration.

**Lemma 24** *For any $k$ and $t \geq 0$, we have*

$$
\sigma_1^2(\boldsymbol{u}_{k,t+1}) \geq (1 + 2\eta\lambda_k - 2\eta\sigma_1^2(\boldsymbol{u}_{k,t}) - 2\eta\sigma_1^2(\boldsymbol{K}_{k,t}\boldsymbol{\Pi}_{\boldsymbol{u}_{k,t}})) \cdot \sigma_1^2(\boldsymbol{u}_{k,t}) - 2\eta\sigma_1^2(\boldsymbol{u}_{k,t}\boldsymbol{G}_{k-1,t}^\top).
$$

**Proof** Substituting (22) into $\sigma_1^2(\boldsymbol{u}_{k,t+1})$ gives that

$$
\begin{aligned}
\sigma_1^2(\boldsymbol{u}_{k,t+1}) &= \boldsymbol{u}_{k,t+1}\boldsymbol{u}_{k,t+1}^\top \\
&= \boldsymbol{u}_{k,t}(\boldsymbol{I}_r + \eta\lambda_k\boldsymbol{I}_r - \eta\boldsymbol{X}_t^\top\boldsymbol{X}_t)^2\boldsymbol{u}_{k,t}^\top \\
&= \boldsymbol{u}_{k,t}(\boldsymbol{I}_r + 2\eta\lambda_k\boldsymbol{I}_r - 2\eta\boldsymbol{X}_t^\top\boldsymbol{X}_t)\boldsymbol{u}_{k,t}^\top + \eta^2\boldsymbol{R}_{k,t} \\
&= \boldsymbol{u}_{k,t}(\boldsymbol{I}_r + 2\eta\lambda_k\boldsymbol{I}_r - 2\eta\sigma_1^2(\boldsymbol{u}_{k,t})\boldsymbol{I}_r - 2\eta\sigma_1^2(\boldsymbol{K}_{k,t}\boldsymbol{\Pi}_{\boldsymbol{u}_k,t})\boldsymbol{I}_r - 2\eta\boldsymbol{G}_{k-1,t}^\top\boldsymbol{G}_{k-1,t})\boldsymbol{u}_{k,t}^\top \\
&\quad + 2\eta\boldsymbol{R}_{k,t}' + \eta^2\boldsymbol{R}_{k,t},
\end{aligned}
$$

where $\boldsymbol{R}_{k,t}$ and $\boldsymbol{R}_{k,t}'$ are non-negative real numbers given by

$$
\begin{aligned}
\boldsymbol{R}_{k,t} &= \boldsymbol{u}_{k,t}(\lambda_k\boldsymbol{I}_r - \boldsymbol{X}_t^\top\boldsymbol{X}_t)^2\boldsymbol{u}_{k,t}^\top, \\
\boldsymbol{R}_{k,t}' &= \boldsymbol{u}_{k,t}(\sigma_1^2(\boldsymbol{K}_{k,t}\boldsymbol{\Pi}_{u_k,t})\boldsymbol{I}_r - \boldsymbol{\Pi}_{\boldsymbol{u}_k,t}\boldsymbol{K}_{k,t}^\top\boldsymbol{K}_{k,t}\boldsymbol{\Pi}_{\boldsymbol{u}_k,t})\boldsymbol{u}_{k,t}^\top.
\end{aligned}
$$

It then follows that

$$
\sigma_1^2(\boldsymbol{u}_{k,t+1}) \geq (1 + 2\eta\lambda_k - 2\eta\sigma_1^2(\boldsymbol{u}_{k,t}) - 2\eta\sigma_1^2(\boldsymbol{K}_{k,t}\boldsymbol{\Pi}_{\boldsymbol{u}_k,t})) \cdot \sigma_1^2(\boldsymbol{u}_{k,t}) - 2\eta\sigma_1^2(\boldsymbol{u}_{k,t}\boldsymbol{G}_{k-1,t}^\top),
$$

which concludes the proof. ∎

Lemma 25 provides an upper bound on $\sigma_1(\boldsymbol{u}_{k,t+1}\boldsymbol{K}_{k,t+1}^\top)$ in terms of the preceding iteration.

**Lemma 25** *Suppose $\eta \leq \frac{1}{12\lambda_1}$ and $\sigma_1^2(\boldsymbol{X}_t) \leq 2\lambda_1$. For any $k \leq r$, if $\sigma_1^2(\boldsymbol{u}_{k,t}) > 0$, then we have*

$$
\begin{aligned}
&\sigma_1(\boldsymbol{u}_{k,t+1}\boldsymbol{K}_{k,t+1}^\top) \\
&\leq \left(1 + \eta\lambda_k + \eta\lambda_{k+1} - 2\eta\sigma_1^2(\boldsymbol{u}_{k,t}) - 2\eta\sigma_1^2(\boldsymbol{K}_{k,t}\boldsymbol{\Pi}_{\boldsymbol{u}_k,t}) + 25\eta^2\lambda_1^2\right) \cdot \sigma_1(\boldsymbol{u}_{k,t}\boldsymbol{K}_{k,t}^\top) \\
&\quad + 3\eta\sigma_1(\boldsymbol{u}_{k,t}\boldsymbol{G}_{k-1,t}^\top)\sigma_1(\boldsymbol{K}_{k,t}\boldsymbol{G}_{k-1,t}^\top).
\end{aligned}
$$

**Proof** Substituting (22) and (23) into $\boldsymbol{u}_{k,t+1}\boldsymbol{K}_{k,t+1}^\top$ gives that

$$
\begin{aligned}
\boldsymbol{u}_{k,t+1}\boldsymbol{K}_{k,t+1}^\top &= \boldsymbol{u}_{k,t}\boldsymbol{K}_{k,t}^\top + \eta\lambda_k\boldsymbol{u}_{k,t}\boldsymbol{K}_{k,t}^\top + \eta\boldsymbol{u}_{k,t}\boldsymbol{K}_{k,t}^\top\boldsymbol{\Gamma}_k - 2\eta\boldsymbol{u}_{k,t}\boldsymbol{X}_t^\top\boldsymbol{X}_t\boldsymbol{K}_{k,t}^\top + \eta^2\boldsymbol{E}, \\
&= \boldsymbol{B} + \boldsymbol{C} - 2\eta\boldsymbol{D} + \eta^2\boldsymbol{E},
\end{aligned}
$$

where

$$
\begin{aligned}
\boldsymbol{B} &= \boldsymbol{u}_{k,t}\boldsymbol{K}_{k,t}^\top\left(\frac{1}{2}\boldsymbol{I}_{d-k} - 2\eta\boldsymbol{K}_{k,t}\boldsymbol{\Pi}_{\boldsymbol{u}_k,t}\boldsymbol{K}_{k,t}^\top\right) \\
\boldsymbol{C} &= \boldsymbol{u}_{k,t}\boldsymbol{K}_{k,t}^\top\left(\frac{1}{2}\boldsymbol{I}_{d-k} + \eta\lambda_k\boldsymbol{I}_{d-k} - 2\eta\sigma_1^2(\boldsymbol{u}_{k,t})\boldsymbol{I}_{d-k} + \eta\boldsymbol{\Gamma}_k - 2\eta\boldsymbol{K}_{k,t}(\boldsymbol{I}_r - \boldsymbol{\Pi}_{\boldsymbol{u}_k,t})\boldsymbol{K}_{k,t}^\top\right), \\
\boldsymbol{D} &= \boldsymbol{u}_{k,t}\boldsymbol{G}_{k-1,t}^\top\boldsymbol{G}_{k-1,t}\boldsymbol{K}_{k,t}^\top, \\
\boldsymbol{E} &= \lambda_k\boldsymbol{u}_{k,t}\boldsymbol{K}_{k,t}^\top\boldsymbol{\Gamma}_k - \boldsymbol{u}_{k,t}\boldsymbol{X}_t^\top\boldsymbol{X}_t\boldsymbol{K}_{k,t}^\top\boldsymbol{\Gamma}_k - \lambda_k\boldsymbol{u}_{k,t}\boldsymbol{X}_t^\top\boldsymbol{X}_t\boldsymbol{K}_{k,t}^\top + \boldsymbol{u}_{k,t}(\boldsymbol{X}_t^\top\boldsymbol{X}_t)^2\boldsymbol{K}_{k,t}^\top.
\end{aligned}
$$

By the singular value inequality,

$$
\sigma_1(\boldsymbol{u}_{k,t+1}\boldsymbol{K}_{k,t+1}^\top) \leq \sigma_1(\boldsymbol{B}) + \sigma_1(\boldsymbol{C}) + 2\eta\sigma_1(\boldsymbol{D}) + \eta^2\sigma_1(\boldsymbol{E}).
$$

For the first term $\boldsymbol{B}$, observe that

$$
\begin{aligned}
(\boldsymbol{u}_{k,t}\boldsymbol{u}_{k,t}^\top)^{-1/2}\boldsymbol{B} &= (\boldsymbol{u}_{k,t}\boldsymbol{u}_{k,t}^\top)^{-1/2}\boldsymbol{u}_{k,t}\boldsymbol{K}_{k,t}^\top\left(\frac{1}{2}\boldsymbol{I}_{d-k} - \boldsymbol{K}_{k,t}\boldsymbol{\Pi}_{\boldsymbol{u}_k,t}\boldsymbol{K}_{k,t}^\top\right) \\
&= \left(1/2 - \sigma_1^2((\boldsymbol{u}_{k,t}\boldsymbol{u}_{k,t}^\top)^{-1/2}\boldsymbol{u}_{k,t}\boldsymbol{K}_{k,t}^\top\right) \cdot (\boldsymbol{u}_{k,t}\boldsymbol{u}_{k,t}^\top)^{-1/2}\boldsymbol{u}_{k,t}\boldsymbol{K}_{k,t}^\top \\
&= \left(1/2 - \sigma_1^2(\boldsymbol{K}_{k,t}\boldsymbol{\Pi}_{\boldsymbol{u}_k,t})\right) \cdot (\boldsymbol{u}_{k,t}\boldsymbol{u}_{k,t}^\top)^{-1/2}\boldsymbol{u}_{k,t}\boldsymbol{K}_{k,t}^\top.
\end{aligned}
$$

where we use the equality $\sigma_1(\boldsymbol{K}_{k,t}\boldsymbol{\Pi}_{\boldsymbol{u}_k,t}) = \sigma_1((\boldsymbol{u}_{k,t}\boldsymbol{u}_{k,t}^\top)^{-1/2}\boldsymbol{u}_{k,t}\boldsymbol{K}_{k,t}^\top)$. Thus,

$$
\sigma_1(\boldsymbol{B}) = (1/2 - \sigma_1^2(\boldsymbol{K}_{k,t}\boldsymbol{\Pi}_{\boldsymbol{u}_k,t})) \cdot \sigma_1(\boldsymbol{u}_{k,t}\boldsymbol{K}_{k,t}^\top).
$$

For the second term $C$, by the singular value inequality,

$$\sigma_1(C)$$
$$\leq \sigma_1\left(\frac{1}{2}I_{d-k} + \eta\lambda_k I_{d-k} - 2\eta\sigma_1^2(u_{k,t})I_{d-r} + \eta\Gamma_k - 2\eta K_{k,t}(I_r - \Pi_{u_k,t})K_{k,t}^\top\right) \cdot \sigma_1(u_{k,t}K_{k,t}^\top)$$
$$\leq (1/2 + \eta\lambda_k - 2\eta\sigma_1^2(u_{k,t}) + \eta\lambda_{k+1}) \cdot \sigma_1(u_{k,t}K_{k,t}^\top).$$

For the third term $D$, $\sigma_1(D) \leq \sigma_1(u_{k,t}G_{k-1,t}^\top)\sigma_1(K_{k,t}G_{k-1,t}^\top)$. For the fourth term $E$, since $\sigma_1^2(X_t) \leq 2\lambda_1$, we have

$$\sigma_1(E) \leq 25\lambda_1^2\sigma_1(u_{k,t}K_{k,t}) + 8\lambda_1\sigma_1(u_{k,t}G_{k-1,t}^\top)\sigma_1(K_{k,t}G_{k-1,t}^\top).$$

Combining, we prove the lemma. ∎

Lemma 26 provides an upper bound on a specific ratio, which is used in the proof of Lemma 22. It serves as a new variant of the SNR argument.

**Lemma 26** *Suppose* $\eta \leq \frac{\Delta}{100\lambda_1^2}$, $\sigma_1^2(X_t) \leq 2\lambda_1$, *and* $\lambda_{k+1} \leq \lambda_k - \Delta$. *Let*

$$\text{ratio} := \frac{1 + \eta\lambda_k + \eta\lambda_{k+1} - 2\eta\sigma_1^2(u_{k,t}) - 2\eta\sigma_1^2(K_{k,t}\Pi_{u_k,t}) + 25\eta^2\lambda_1^2}{1 + 2\eta\lambda_k - \eta\Delta/4 - 2\eta\sigma_1^2(u_{k,t}) - 2\eta\sigma_1^2(K_{k,t}\Pi_{u_k,t})}.$$

*Then* ratio $\leq 1 - \eta\Delta/6$.

**Proof** Since $\eta \leq \frac{\Delta}{100\lambda_1^2}$ and $\lambda_{k+1} < \lambda_k - \Delta$, we have

$$\text{ratio} \leq \frac{1 - \eta\Delta/4 + s_0}{1 + \eta\Delta/4 + s_0},$$

where

$$s_0 = 2\eta\lambda_k - \eta\Delta/2 - 2\eta\sigma_1^2(u_{k,t}) - 2\eta\sigma_1^2(K_{k,t}\Pi_{k,t}) \in [-1/2, 1/2].$$

Since the function $g_9(s) = \frac{1-\eta\Delta/4+s}{1+\eta\Delta/4+s}$ is increasing on $[-1/2, 1/2]$, we have

$$\text{ratio} \leq \frac{1 - \eta\Delta/4 + 1/2}{1 + \eta\Delta/4 + 1/2} \leq 1 - \eta\Delta/6,$$

which concludes the proof. ∎

## C  PROOF OF PROPOSITION 7

**Proof** Consider $X$ with $\sigma_1(X) \leq \frac{1}{\sqrt{3\eta}}$. Let $X_t$ be the GD sequence initialized by $X$. By Corollary 2 of Lee et al. (2019), we know GD sequence almost surely avoids the strict saddle points. By Zhu et al. (2021), we know all the saddle points are strict and all the local minima are global minima. Therefore, we conclude that the GD sequence converges to the global minima almost surely.

Now it remains to show that Assumption 4 must hold if the GD sequence converges to the global minima. Indeed, if we suppose Assumption 4 does not hold, then the GD sequence will converge with $\lim_{t\to\infty} \sigma_1(u_{k,t}) = 0$ for some $k \leq r$. This means the GD sequence converges to a saddle point, since any stationary point with some $u_{k,t} = 0$ ($k \leq r$) is a saddle point, rather than a global minimum. This leads to the contradiction. ∎

## D  ADDITIONAL EXPERIMENTS

In this section provide additional experiments to support and illustrate our theoretical results.

## D.1 RANK-TWO MATRIX APPROXIMATION

Our first extended experiment examines rank-two matrix approximation with varying dimension $d$ and initial magnitude $\varpi$. Specifically, we will choose $d$ from the set $\{1000, 2000, 4000\}$ and choose $\varpi$ from the set $\{0.001, 0.5, 2\}$. For each $d$, we set $\boldsymbol{\Sigma} = \mathrm{diag}(\boldsymbol{a}, \boldsymbol{e})$, where $\boldsymbol{a} \in \mathbb{R}^r$ is a decreasing arithmetic sequence starting from 1 to 0.5 and $\boldsymbol{e} \in \mathbb{R}^{d-r}$ is an arithmetic sequence transitioning from 0.3 to zero. Let $\boldsymbol{X}_0 = \varpi \boldsymbol{N}_0$ with the entries of $\boldsymbol{N}_0$ independently drawn from $\mathcal{N}(0, \frac{1}{d})$. We compute the GD sequence $\boldsymbol{X}_t$ with a step size of 0.1 and evaluate the errors $\|\boldsymbol{\Sigma}_r - \boldsymbol{X}_t\boldsymbol{X}_t^\top\|_\mathrm{F}$, where $\boldsymbol{\Sigma}_r = \mathrm{diag}(\boldsymbol{a}, \boldsymbol{0})$ is the best rank-r approximation to $\boldsymbol{\Sigma}$. The error curves of GD for different settings are displayed in Figure 2.

Figure 2 demonstrate that all the error curves exhibit the similar behaviors. The only differences lie on the first stage.

- When we use a small $\varpi = 0.001$, the error does not rapidly change at the beginning. This is because $\|\boldsymbol{X}_t\|_\mathrm{F}$ is close to zero and the error $\|\boldsymbol{\Sigma}_r - \boldsymbol{X}_t\boldsymbol{X}_t\|_\mathrm{F}$ is approximately $\|\boldsymbol{\Sigma}_r\|_\mathrm{F}$. This period of time corresponds to the second property of Theorem 6.

- When we use $\varpi = 2$, we find the error first drops rapidly from a large value to $\|\boldsymbol{\Sigma}_r\|$. This corresponds to the Lemma 8 and the first property in Theorem 6.

- When we use $\varpi = 0.5$, the first stage nearly disappears. This means that $T_{\boldsymbol{u}_1}$ in Theorem 6 is small, especially compared with the case where $\varpi = 0.001$.

In addition, we want to mention that if we use $\varpi = 10$ to initialize the algorithm and keep other settings unchanged, then the GD sequence will diverge. This serves as a supplementary to the above experimental results.

## D.2 GENERAL RANK MATRIX APPROXIMATION

Our second experiment examines general rank matrix approximation, where we fix dimension $d = 1000$ and vary the rank $r$ across $\{2, 6, 10\}$. In addition, for each setting, we examine different initial magnitudes $\varpi \in \{0.001, 0.5, 2\}$. Our setting for $\boldsymbol{\Sigma}$ is the same as before, that is, $\boldsymbol{\Sigma} = \mathrm{diag}(\boldsymbol{a}, \boldsymbol{e})$ with $\boldsymbol{a} \in \mathbb{R}^r$ and $\boldsymbol{e}^{d-r}$ being two arithmetic sequences. We initialize GD using $\boldsymbol{x}_0 = \varpi \boldsymbol{N}_0$ and we compute the GD sequence and the errors $\|\boldsymbol{\Sigma}_r - \boldsymbol{X}_t\boldsymbol{X}_t\|_\mathrm{F}$. The results are displayed in Figure 3.

As the results demonstrate, the effects of $\varpi$ is similar to the one in Section D.1. Moreover, we observe another interesting phenomenon that may need additional explanations. Figure 3 shows that the error curve for larger rank $r$ is smoother than the one for smaller rank $r$. Our explanation is that for larger rank $r$, the differences between successive eigenvalues are smaller. Thus, it is harder to distinguish the associated eigenvectors, and all the eigenvectors may be learned together. As a result, the error curve remains decreasing along the iterations.

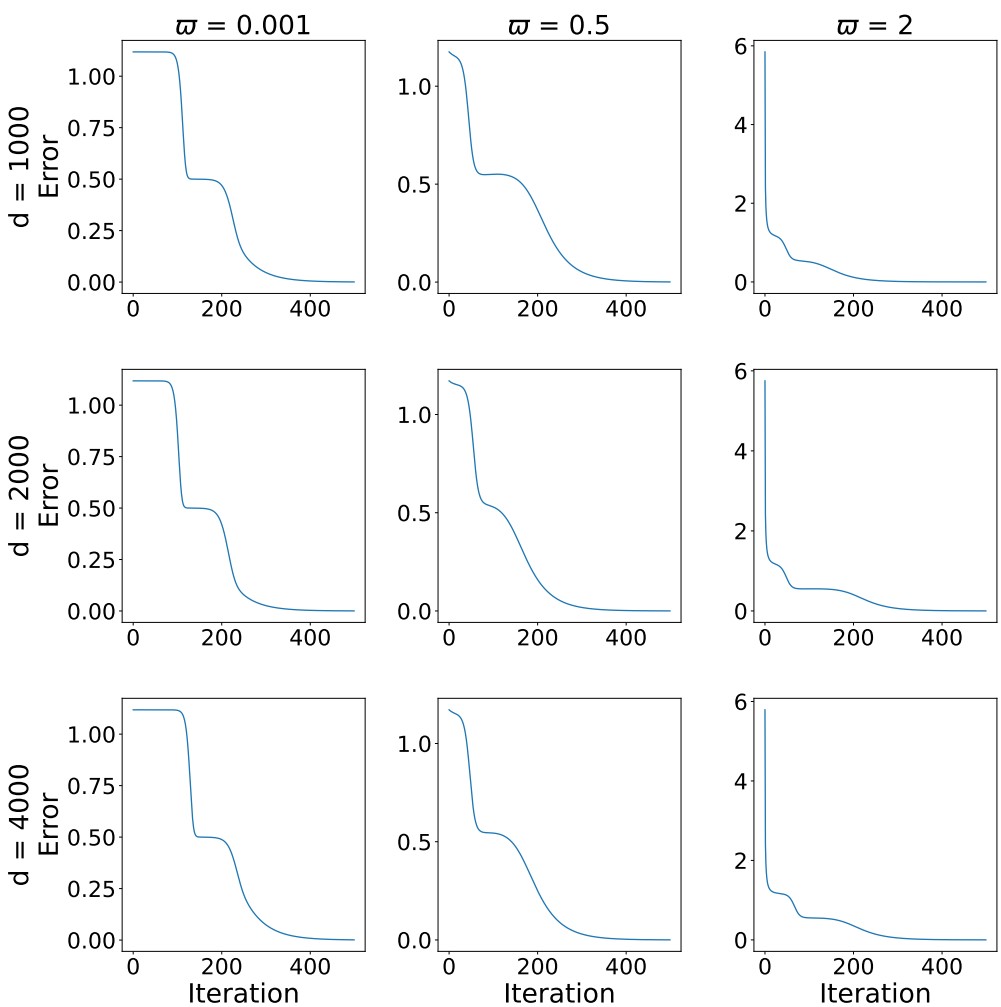

Figure 2: Error curves of GD, measured by $\|\mathbf{\Sigma}_r - \mathbf{X}_t \mathbf{X}_t^\top\|_F$, for rank-two matrix approximation. The columns represent different initial magnitudes $\varpi = 0.001, 0.5, 2$. The rows represent different dimensions $d = 1000, 2000, 4000$.

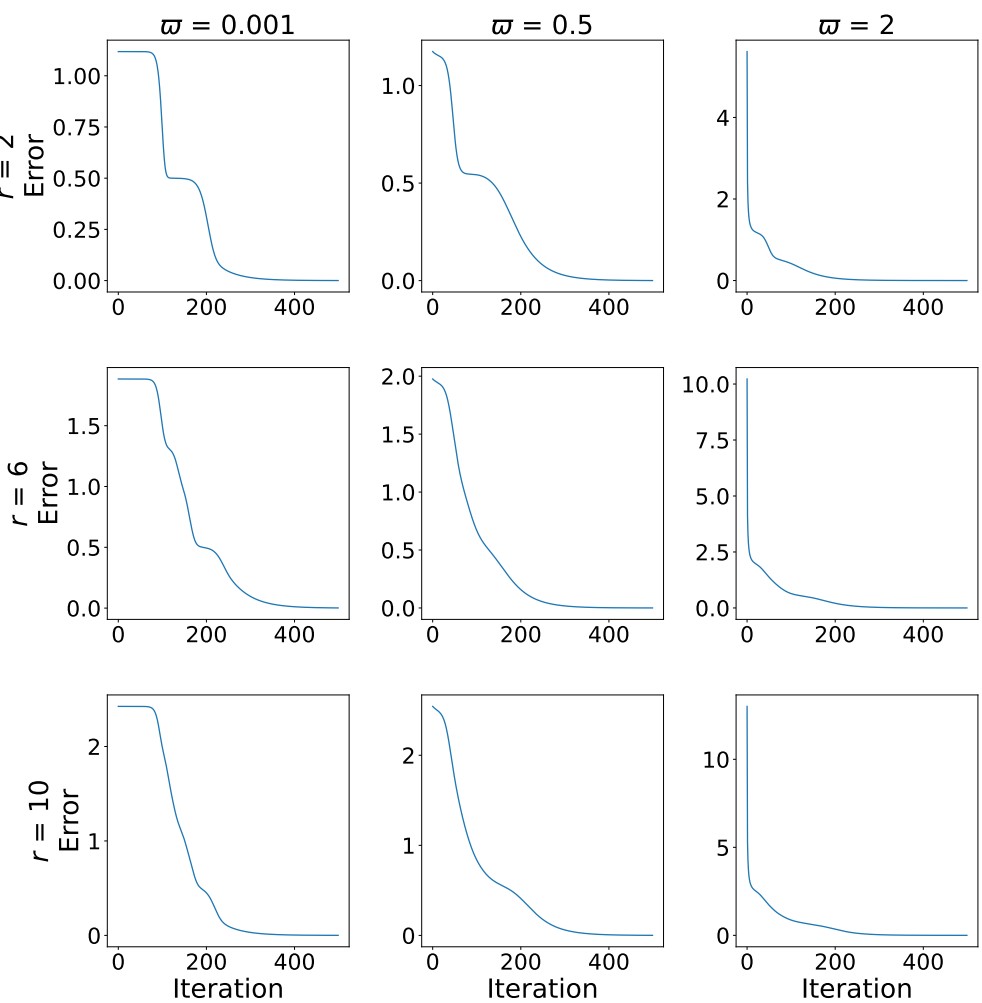

Figure 3: Error curves of GD, measured by $\|\mathbf{\Sigma}_r - \boldsymbol{X}_t \boldsymbol{X}_t^\top\|_{\mathrm{F}}$, for general rank matrix approximation. The dimension $d$ is set as 1000. Different rows represent different rank $r$. Different columns represent different initial magnitudes $\varpi$.