# OpenReview forum: "Gradient descent for matrix factorization: Understanding large initialization"
_ICLR.cc/2024/Conference — Submitted to ICLR 2024_

### Official Review · Reviewer_MbHg · 2023-10-27

**Soundness:** 2 fair
**Presentation:** 2 fair
**Contribution:** 3 good
**Rating:** 6
**Confidence:** 3

**Summary:**

This paper analyzes the incremental learning behavior of gradient descent for matrix factorization problems. Different from existing works that only consider small initialization, this paper considers initializations that have constant scale. Under certain regularity assumptions on the initialization and singular values in the training process, the authors show that gradient descent still exhibits incremental learning. In their theoretical analysis, the auhtors use a  novel signal-to-noise ratio (SNR) argument to show the exponential separation between a low-rank signal and a noise term, which allows that to show that gradient descent trajectory is nearly low-rank.

**Strengths:**

1) The introduction part of this paper is written in a clear and succint manner. The main theoretical results are often followed by necessary explanations and proof sketch, making it easier for the readers to understand them.
2) The effect of large initialization on implicit bias is an interesting topic and the theoretical results in this paper are novel to the best of my knowledge. The authors also provide comprehensive review of the related literature.

**Weaknesses:**

1) The theoretical results in this paper are presented in a somewhat isolated manner, and it seems that then result for rank-$2$ is not even stated as an independent theorem. I suggest that the authors can briefly summarize the main results (for rank-$2$ and general ranks) in the introduction, before going into technical details.

2) The organization of Sec. 3 can probably be improved: although the title of this section is "challenges in examine general rank solutions", Sec. 3.1 is about local convergence and the remaining two subsections seem to discuss the challenges for large initialization, rather than for general ranks.

3) The "signal-noise-ratio" argument in this work looks different from existing works that also decompose the GD trajectory in the signal and noise term (e.g. [1,2] ), but the decomposition seems to be the same. I suggest that the authors can add more discussions about this difference to highlight the contribution of this paper.

[1] Stoger, D. and Soltanolkotabi, M. Small random initialization is akin to spectral learning: Optimization and generalization guarantees for overparameterized low-rank matrix reconstruction.
[2] Jin, J., Li, Z., Lyu, K., Du, S. S., & Lee, J. D. Understanding incremental learning of gradient descent: A fine-grained analysis of matrix sensing.

**Questions:**

1) I am confused about Assumption 8 and cannot see how it is related to the arguments at the beginning of Sec. 4.2.3. According to my understanding, it says that the norm of the $2$-to-$d$ components is larger than its inner product with the first component. Can you explain this assumption in more details?

2) The authors seem to employ a successive argument for general rank matrix. However, the definition of the benign set $R$ in Sec. 3.1 would change for higher ranks. Is there any arguments in your proof verifying that GD remains in the high-rank $R$? Probably you did it in Theorem 13, but I cannot understand how they are related.

3) In existing works it is commonly the case that convergence/incremental learning hold with high probability (e.g.[1] Theorem 3.3), since the initialization has to be aligned with each component, otherwise it cannot make progess in some direction. Does this paper need to impose similar requirements for initialization?

[1] Stoger, D. and Soltanolkotabi, M. Small random initialization is akin to spectral learning: Optimization and generalization guarantees for overparameterized low-rank matrix reconstruction.

---

> ### Author Response · Authors · 2023-11-16
> **Rebuttal for Reviewer MbHg - Part one**
>
> Thanks the reviewer for the valuable suggestions and questions. In this rebuttal, we will address them point by point. Due to the limitation of characters, we separate this rebuttal into two parts. The first one addresses the weakness section. The second one addresses the remaining questions.
>
> **Weakness**
>
> - The theoretical results in this paper are presented in a somewhat isolated manner, and it seems that then result for rank-2 is not even stated as an independent theorem. I suggest that the authors can briefly summarize the main results (for rank-2 and general ranks) in the introduction, before going into technical details.
>     - Thanks for the reviewer’s suggestion. In our updated version, we summarize our results in a main theorem, Theorem 6, and an intuitive theorem, Theorem 1. We put the intuitive theorem in the introduction. We also move the visualization results and the intuitive ideas to the introduction. We move the analysis of rank-2 results to a later section called sketch of proof.
> - The organization of Sec. 3 can probably be improved: although the title of this section is "challenges in examine general rank solutions", Sec. 3.1 is about local convergence and the remaining two subsections seem to discuss the challenges for large initialization, rather than for general ranks
>     - Thanks for the reviewer’s suggestion. We have changed the structure for that section. We have made the local linear convergence a single section, emphasizing the usage of SNR in this application. Then we discuss the random initialization. We both review small random initialization and present the main theorem for large initialization. Explanations and comparisons are given in corresponding places. After this, we give a detailed sketch of proof.
> - The "signal-noise-ratio" argument in this work looks different from existing works that also decompose the GD trajectory in the signal and noise term (e.g. [1,2] ), but the decomposition seems to be the same. I suggest that the authors can add more discussions about this difference to highlight the contribution of this paper.
>     - Thanks for the reviewer’s suggestion. We have added more discussions in our updated version. Specifically, to employ the SNR analysis, we need a lower bound of the signal and an upper bound on the noise, both in terms of their previous iterations. These two bounds need to be related so that the ratio of SNR$_{t+1}$  by SNR$_t$ can be analyzed. The challenging part is to obtain two related upper and lower bounds. Previous literature does not provide SNR considerations so their upper and lower bounds are not applicable for our purpose. In addition, for the large initialization setting, our SNR analysis is even more novel, because we introduce new signal and noise terms.
>
> Hope this addresses the reviewer's concerns regarding the weaknesses.

---

> > ### Author Response · Authors · 2023-11-16
> > **Rebuttal for Reviewer MbHg - Part two**
> >
> > We would like to thank the reviewer for the valuable suggestions and questions again. In this part, we address the reviewer's questtions.
> >
> > - **Questions**
> > - I am confused about Assumption 8 and cannot see how it is related to the arguments at the beginning of Sec. 4.2.3. According to my understanding, it says that the norm of the 2-to-infinity components is larger than its inner product with the first component. Can you explain this assumption in more details
> >     - Thanks for the question. First, the reviewer may misunderstand the meaning of this assumption. In this assumption, we assume the existence of $t_1^*$ such that
> >
> >         $r(1-\eta\Delta/6)^{t_{1}^*}\leq\sqrt{\frac{\Delta}{8}}\min\{\sigma_1(u_{2,t_1+t_1^*}),\sqrt{\frac{\Delta}{2}}\}$
> >
> >         Recall that for all $t\geq t_1$, we have shown that
> >         $\sigma_1(u_{1,t}K_{1,t}^\top)\leq(1-\eta\Delta/6)^{t-t_{1}}$
> >
> >         Therefore, the condition for $t_1^*$ means the second signal $\sigma_1(u_{2,t_1+t_1^*})$ is larger than the noise term $\sigma_1(u_{1,t}K_{1,t}^\top).$ Hence, when we analyze the second signal $\sigma_1(u_{2,\cdot})$, we could disregard the effect of the noise term $u_{1,t}K_{1,t}^\top$. This allows us to use the inductive argument even when the noise term is not exactly zero.  We have also added this explanation to the paper in our updated version.
> >
> > - The authors seem to employ a successive argument for general rank matrix. However, the definition of the benign set in Sec. 3.1 would change for higher ranks. Is there any arguments in your proof verifying that GD remains in the high-rank? Probably you did it in Theorem 13, but I cannot understand how they are related.
> >     - Thanks for the question. We are afraid that the reviewer may misunderstand some points here. Let us clarify for the reviewer and answer all the questions.
> >     - When we want to study rank-r approximation, we will fix the benign set in Sec 3.1 as a rank-r set and we never change it. To apply this result in Theorem, we need to prove that after r inductive arguments, the GD sequence will enter the benign set defined in Sec 3.1. This is because when all r signals are strong enough and noise are weak, the sequence will enter the desired strong SNR region.
> >     - Consider the heat plot of $XX^\top$ and the first $r\times r$ block. Our argument is to show that the first r diagonal elements will increase above $\Delta/2$ and the norm of all off-diagonal elements in this block decrease fast to zero. Combining, we know this block’s smallest singular value is larger than  $\Delta/4$.
> >     - The challenge is to verify that all r signals can be strong enough and noise are weak. We prove this using our novel inductive argument.
> >     - In our updated version, we have clearly discussed the relationship between local linear convergence theorem and the theorem using large initialization. Hope this has addressed the reviewer's question.
> > - In existing works it is commonly the case that convergence/incremental learning hold with high probability (e.g.[1] Theorem 3.3), since the initialization has to be aligned with each component, otherwise it cannot make progress in some direction. Does this paper need to impose similar requirements for initialization?
> >     - Yes, if we employ random initialization, our results apply to them with high probability. This is the best outcome we can hope because it is always possible that the initialization is pretty bad so that the algorithm can diverge. For example, a extremely large $\sigma_1(X_0)$ could result in the divergence of GD.
> >     - In addition, we would like to mention the different requirements between ours and [1]’s. The main difference is the initialization magnitude. They require small initialization, and specifically $\varpi \leq  \min(d^{-1/2},d^{-3\kappa^2})$, where $\kappa>1$ is the condition number. In contrast, we only require $\varpi=O(1/\sqrt{\eta})$ where $\eta$ is the step size. Notably, our requirement is independent of d. Also, our requirement is rate-optimal.
> >
> > If the reviewer has additional questions, please feel free to let us know.

---

> > > ### Comment · Reviewer_MbHg · 2023-11-16
> > > **Reply to the rebuttal**
> > >
> > > I would like to thank the authors for clarification, and I think I can now understand the idea of your proof.
> > >
> > > Now it seems to me that your proof of convergence rely on the assumption that $r(1-\eta \Delta / 6)^{t_1^*} \leq \sqrt{\frac{\Delta}{8}} \min \sigma_1\left(u_{2, t_1+t_1^*}\right), \sqrt{\frac{\Delta}{2}}$, so that you would have a "good starting point" for the second round. Do you think it is possible to remove this algorithm-dependent assumption, or verifying it with simple experiments? Recent works with smaller init should be able to address this issue.

---

> > > > ### Author Response · Authors · 2023-11-16
> > > > **Response to the follow-up question**
> > > >
> > > > We would like to thanks for the reviewer's response. Here is our response to the follow-up question.
> > > >
> > > > - In our updated version, we distinguish two different assumptions as follows. First, we define $t_1^*$ as the smallest integer such that
> > > > $$r(1-\eta\Delta/6)^{t_{1}^*}\leq\sqrt{\frac{\Delta}{8}}\min(\sigma_1(u_{2,t_1+t_1^*}),\sqrt{\frac{\Delta}{2}}).$$
> > > > **The first assumption is that** $t_1^*<\infty$ **and the second assumption is that** $t_1^*=O(\log(d))$. **The second assumption is clearly a stronger one**. In our updated version, these assumptions are Assumption 4 and 5 on page 6 respectively.
> > > >
> > > > - **Our trajectory analysis remains valid when only the first assumption** ($t_1^*<\infty$) **is used**. In other words, the first assumption is already sufficient for establishing properties 1-6 in our main theorem, Theorem 6 on page 6. These results already explain the incremental learning phenomenon. Meanwhile, **we have demonstrated that the first assumption holds almost surely if we employ random initialization**. In fact, if this assumption does not hold, it means the second signal $\sigma_1(u_{2,t})$ will converge to zero. This implies that the GD sequence converge to a saddle point. However, this almost never happens if we use random initialization. The previous literature has demonstrated that for the matrix factorization problem, GD with random initialization almost surely converges to the global minimum rather than saddle points. Therefore, **we could say this first assumption is verified**. We need it only because our result is a deterministic one, rather than assuming random initialization.
> > > >
> > > > - **The second assumption** ($t_1^*=O(\log(d))$) **is only needed for the seventh property** in our main theorem. We need this assumption to show that GD achieves $\epsilon$-accuracy in $O(\log(d)+\log(1/\epsilon))$ iterations. **It is theoretically challenging to verify this second assumption, which in some sense reflects the difficulty of our considered problem. In our paper, we have provided many discussions related to this assumption for additional insights.**
> > > >     - First, one can treat small initialization as a special setting of our theorem, where the previous literature have demonstrate the second assumption. However, our theorem also applies to large initialization setting which is commonly used in practice. Therefore, our theorem is a strict generalization of previous results, at least for the matrix factorization problem.
> > > >     - Second, this assumption intuitively compares the noise term $\sigma_1(u_{1,t}K_{1,t}^\top)$ and the signal term $\sigma_1(u_{2,t})$. We theoretically prove the the noise term decreases to zero geometrically fast, which relates to the geometrically decaying sequence $r(1-\eta\Delta/6)^{t}$. However, there is no such theory saying the signal term $\sigma_1(u_{2,t})$ would also decay geometrically fast to zero. Therefore, we may expect this comparison is informative. Also, we may expect the assumption $t_2^*=O(\log(d))$ may hold.
> > > >     - Third, our experiments have demonstrated that this assumption holds. In fact, if GD converges to the global minimum geometrically fast, then this assumption automatically holds. The fast global convergence of GD for matrix factorization problems have been observed long in our experiments and many other related works.
> > > >     - Fourth, if this assumption does not hold, then the statement that GD converges fast to the global minimum could be wrong. Thus, it could be a starting point to establish some negative results. After all, no one has theoretically studied GD with large initialization before, and all the experimental results could be misleading. It is beneficial to doubt whether the conjecture of fast convergence is right or not.
> > > >     - Finally, although previous results have studied small initialization, as we mentioned, their assumptions could be too strong. The initialization scale $\varpi\lesssim \min (d^{-1/2},d^{-3\kappa^2})$ is rarely used in practice. Therefore, it motivates us to study a practical setting, i.e., the large initialization regime. Although we do not offer a fully complete result due to the challenging difficulties, we still use novel analysis to derive many meaningful results and insights. We do hope the reviewer could appreciate that.

---

> > > > > ### Comment · Reviewer_MbHg · 2023-11-16
> > > > > **Response to the rebuttal**
> > > > >
> > > > > I would like to thank the authors for clarifications.
> > > > >
> > > > > Now I think I can fully understand the contributions of this paper, though it might be better to state that "assumption 4 almost surely holds" as a separate proposition and include a formal proof. I will increase my rating to 6.

---

> > > > > > ### Author Response · Authors · 2023-11-17
> > > > > > **Response**
> > > > > >
> > > > > > Thanks for increasing the rating.
> > > > > >
> > > > > > We have included a new proposition with a formal proof in our latest version of the paper.

---

### Official Review · Reviewer_BGNj · 2023-10-31

**Soundness:** 2 fair
**Presentation:** 2 fair
**Contribution:** 2 fair
**Rating:** 5
**Confidence:** 4

**Summary:**

The paper studies the influence of initialization on the performance of gradient descent for the matrix factorization problems.
Existing work focusses on small initialization (see missing ref in References).
The analysis is based on a signal-to-noise (SNR) analysis, which generalizes the one of CHen et al (2019) for the rank 1 matrix factorization problem. Here the SNR is defined as the ratio of the norm of the components of X aligned with the desired directions (first k eigenspaces of target matrix $\Sigma$), divided by the norm of the remaining components.

The paper has two main contributions:
- Provided GD is initialized in a high SNR region (Eq 10), the authors prove linear convergence towards the global minimizer.
- When no such initialization is available, the paper provides additional results, but (see first point below) it still seems to require that $X_0$ be in certain region which is unknown.

**Strengths:**

Potential important progress in the field of implicit bias for matrix factorization, if the results are true. All previous results I know of are for small initialization.

**Weaknesses:**

See questions below:
- only one experiment, on rank 2 matrix approximation
- unclear dependency on initialization: $\alpha$ needs to be small
- unclear it it is required in the proof that $X_0 \in S$ or not (contradictory statements in the paper)

**Questions:**

### Mathematics
By order of importance:
- The paper claims to study large initialization, and that the first result with initialization in the region of Eq 10 is not satisfying. However, the proof of the main result states "This is achievable if we use random initialization X0 = αN0 with a reasonably small constant α", which is not "large initialization". $\alpha$ must be small enough such that $\sigma^2_1(K_r) \leq \lambda_r - 3 \Delta/4$, so $\alpha$ depends on $r$, and it's not so wild to think that $r$ is a fraction of $d$. Remark 4 seems to contradict that, highlighting that the theorem does not require $X_0 \in S$, but the exposition is very confusing here (eg, Lemma 5 assumes that $X_0 \in S$)
- It is clear that Sigma can be assumed diagonal in problem 1 up to an orthogonal change of variable in $X$ (because GD is equivariant to such a change of variable), but this deserves to be stated explicitly the first time this assumption is made (P2).
- $\sigma_1(u_{k,t})$ is just its norm, right? $u_{k,t}$ is a vector. Same for $u_{k,t} K_{k, t}^\top$


## Experiments
- In the experimental example, how do we know if 0.5 N(0, 1/d) is a large initialization? Since $d$ does not vary, what can we say? It would be better to have the experiment for several values of $d$ and fixed $\bar \omega$
- The experimental example considers rank 2 matrix factorization which, though by nature different as the authors have explained, does not seem to far from rank 1 factorization (especially compared to the dimension 2000). It would highlight the goal of the paper better to use something like rank 10 matrix factorization here.
- Please provide more than one experiment in dimension 2. The analysis is incomplete without more proof that the results hold for varying r and d, fixed $\bar \omega$.



### Formulation
- Incremental learning has many possible meanings, it would be nice to clarify it here. Same for "spectral method", I think the amount of details given in section 3.2 should be slightly increased.
- P3 "because at the global minima"/"the set R contains all the global minima": it seems to me that there is a single global minima, the rank-$r$ truncated SVD of $\Sigma$. can the authors clarify?
- First sentence of Section 3.3 is a repetition from above.
- Paragraph 4.1 consider replacing rank-$r$ by rank-2 for clarity (and all instances of $r$ in that paragraph)
- Legend of Figure 1: "top three rows" are the first three rows?

### References:
- I believe the paper is missing the seminal reference "S. Gunasekar, B. E. Woodworth, S. Bhojanapalli, B. Neyshabur, and N. Srebro, “Implicit regularization
in matrix factorization,” in Advances in Neural Information Processing Systems, 2017" which conjectured global convergence of GD to the minimal nuclear norm solution in the case of small initialization.


Typos:
- takes infinity at 0
- the rest elements: the remaining elements/the rest of the elements (several occurrences)

---

> ### Author Response · Authors · 2023-11-16
> **Rebuttal for Reviewer BGNj - Part one**
>
> We thank the reviewer for the valuable suggestions and questions. In this rebuttal, we will address them point by point. Due to the limitation of characters, we separate this rebuttal into two parts. This part responds to the mathematics.
>
> - **Mathematics**
> - The paper claims to study large initialization, and that the first result with initialization in the region of Eq 10 is not satisfying…. However, the proof of the main result states "This is achievable if we use random initialization X0 = αN0 with a reasonably small constant α", which is not "large initialization".  must be small enough such that , so  depends on , and it's not so wild to think that  is a fraction of . Remark 4 seems to contradict that, highlighting that the theorem does not require , but the exposition is very confusing here (eg, Lemma 5 assumes that )
>     - Sorry for the confusion here. In our updated version, we have removed all the confusion part. Specifically, for every results we display, we only require $\sigma_1(X_0)\leq1/\sqrt{3\eta}$ and correspondingly $\varpi\lesssim 1/\sqrt{\eta}$. All the subsequent results are then proved, rather than assumed. In addition, we emphasize that this requirement is rate-optimal because if $\sigma_1(X_0)$ is too large, then the GD sequence can simply diverge. A simple counterexample is when $\Sigma=0$ and $\sigma_1^2(X_0)\geq 3/\eta$. An inductive argument implies that $\sigma_1(X_{t+1})\geq 2\sigma_1(X_t)$  for all t, and thus $X_t$ diverges. We have added this example and explanation below Theorem 6 in our updated version.
>     - We would also like to emphasize the difference between large initialization and small initialization. Large initialization means the norm of $X_0$ converges to a positive constant while small initialization means the norm of $X_0$ converges to zero. By concentration results, the norm of $X_0=\varpi N_0$ is O($\varpi$). Therefore, taking a positive constant $\varpi$ means the large initialization. In contrast, in previous literature that assumes small initialization, they require $\varpi$ to be as small as $\min(d^{-1/2},d^{-3\kappa^2})$, where $\kappa>1$ is the condition number. Therefore, there should be a significant difference between two settings.
> - It is clear that Sigma can be assumed diagonal in problem 1 up to an orthogonal change of variable in  (because GD is equivariant to such a change of variable), but this deserves to be stated explicitly the first time this assumption is made (P2).
>     - Yes. In our revised version, we have mentioned that such assumption is made without loss of generality, and explained the reason.
> - $\sigma_1(u_{1,t})$ is just its norm, right? Same for $u_{k,t}K_{k,t}^\top$.
>     - Yes. Both the norm notation and the singular value notation can be used. We chose the singular value notation for its consistency with other results in the paper.

---

> ### Author Response · Authors · 2023-11-16
> **Rebuttal for Reviewer BGNj - Part two**
>
> We would like to thank the reviewer for the valuable suggestions and questions again. In this part, we will address the remaining issues, including experiments, formulations, reference and typos.
>
> - **Experiment**
> - In the experimental example, how do we know if 0.5 N(0, 1/d) is a large initialization? Since d does not vary, what can we say? It would be better to have the experiment for several values of d and $\varpi$
>     - We appreciate your attention to the details of our matrix factorization experiment. Theoretically, by the concentration results, the norm of $X_t$ is around 0.5. This is comparable to the largest eigenvalue 1 of $\Sigma$. Consequently, the higher order term $X_tX_t^\top X_t$ cannot be disregarded in the GD iterations. In this context, the theory of small initialization fails to account for the convergence behavior observed in GD.
>     - Additionally, we add new experiments with varying d and $\varpi$ in the appendix. In particular, we choose d from 1000, 2000, 4000, and we take $\varpi$ from 0.001, 0.5, 2, and 10. Our findings show that if $\varpi$ is as large as 10 in this setting, the GD sequence will diverge. For the remaining three choices of $\varpi$, we can observe similar convergence behaviors of GD.
> - The experimental example considers rank 2 matrix factorization which, though by nature different as the authors have explained, does not seem to far from rank 1 factorization (especially compared to the dimension 2000). It would highlight the goal of the paper better to use something like rank 10 matrix factorization here.
>     - We appreciate your attention to the details of our matrix factorization experiment. Our experiment is designed to demonstrate our core ideas in an intuitive and visually interpretable manner. By using rank-two matrix factorization setting, selecting five noteworthy points, and comparing their heat plots, we could succinctly illustrate the intuitions of our analysis.
>     - Additionally, we add new experiments to address the reviewer’s question. In the second experiment in our appendix, we examine the setting of fixed dimension d with varying rank r and $\varpi$. We let d=1000 and choose r from {2,6,10} and $\varpi$ from {0.001, 0.5, 2}. The error curves of GD are plotted for these different settings. We find that these results are consistent with the results we have presented before.
> - Please provide more than one experiment in dimension 2. The analysis is incomplete without more proof that the results hold for varying r and d, fixed $\varpi$.
>     - Thanks for the reviewer’s suggestion. In our updated version, we have provided additional experiments in the appendix. For rank-two matrix approximation, we vary both dimension d and the initial magnitude $\varpi$. Our results are consistent with the results we have presented.
>
> - **Formulation**
> - Incremental learning has many possible meanings, it would be nice to clarify it here.
>     - Thank you for the suggestions. We have explained that in our context, incremental learning refers to the process where eigenvectors associated with larger eigenvalues are learned first.
> - Same for "spectral method", I think the amount of details given in section 3.2 should be slightly increased.
>     - We have explained the spectral method. It also means the power method. For these kinds of methods, the eigenvectors related to larger eigenvalues are learned first. Notably, $\sigma_r(U)$ increases faster than $\sigma_1(J)$.
>     - We have extended the content of that section. We add more details on two aspects: the theoretical intuition and their assumptions on small initialization.
> - P3 "because at the global minima"/"the set R contains all the global minima": it seems to me that there is a single global minima, the rank-truncated SVD of $\Sigma$. can the authors clarify?
>     - The global minima refer to all $X$ such that $XX^\top=\Sigma_r.$  While  $\Sigma_r$ is unique, there are multiple $X$ satisfying the equality. Since R is a set of X, it is more accurate to say that the set R contains all the global minima.
> - First sentence of Section 3.3 is a repetition from above. Paragraph 4.1 consider replacing rank- by rank-2 for clarity (and all instances of  in that paragraph)
>     - Yes, we have made all the modifications.
> - Legend of Figure 1: "top three rows" are the first three rows?
>     - Yes.
> - **Reference and Typos**
>     - We added the mentioned reference and revised the typos in our updated version.
>
> If the reviewer has additional concerns, please feel free to let us know.

---

> > ### Author Response · Authors · 2023-11-20
> > **Rebuttal discussion**
> >
> > Dear reviewer,
> >
> > We hope this message finds you well. We are writing to kindly follow up on the rebuttal we submitted for our paper. We have addressed the concerns and suggestions you raised during the review process, including mathematics, experiments, formulations, references, and typos. We would greatly appreciate your feedback on the revisions we've made.  We are eager to hear your thoughts on the updated manuscript and answer any questions you might have after reacting to our response. Please let us know if you have any questions.

---

> ### Author Response · Authors · 2023-11-21
> **Rebuttal discussion**
>
> Hi Reviewer BGNj, We are writing again in hopes that you will respond to our rebuttal and let us know your current concerns so that we could address them. For now, we believe we have addressed all the weaknesses you has mentioned. Specially,
>
> - unclear dependency on initialization: $\varpi$ needs to be small;
>    - we have demonstrated that our initialization condition is dimension-independent and rate-optimal; in particular, we only require $\varpi\lesssim 1/\sqrt{\eta}$, which is in sharp contrast to the previous 'small' requirements $\varpi\lesssim \min(d^{-1/2},d^{-3\kappa^2})$.
> - unclear it it is required in the proof that $X_0\in S$
>    - we have made it clear that we do not need this condition; if this is the point that you are concerned, then we have demonstrated that such condition can be completely removed; hope this could address the reviewer's concern.
> - only one experiment, on rank 2 matrix approximation
>    - we have added more experiments in the appendix; due to the limitation of pages, we remain a single experiment in the main text for illustrating our ideas.
>
> Please let us know if you have any concerns so that we could further address them. We are looking forward to your reply!

---

### Official Review · Reviewer_yiZg · 2023-10-31

**Soundness:** 3 good
**Presentation:** 2 fair
**Contribution:** 2 fair
**Rating:** 5
**Confidence:** 2

**Summary:**

This paper focuses on a simplified matrix factorization problem, aiming to understand the convergence of gradient descent when using large random initialization.

**Strengths:**

This paper focuses on a simplified matrix factorization problem, aiming at understanding the convergence of GD when using large random initialization.

**Weaknesses:**

The presentation could be further improved.   The authors claim that they aim to understand the convergence of GD with large random initialization on simple matrix factorization problems, but as a reader, I could not find a main theorem or corollary that clearly states that with large random initialization, GD converges with certain rates under appropriate parameter settings and assumptions.  Also, the presentation of the theorem involves many notations, which on the other hand, looks could not be uninvolved.

The  'large initialization' is in fact the `large random initialization'.  The authors consider GD with large random initialization, but the variance still depends on the dimension of the problem under consideration. Therefore, when the dimension d is relatively large, it reduces to the ``small" random initialization setting.

The authors consider a simple matrix factorization problem, but as claimed in the main text, the motivation of this paper is to better understand  GD with large random initialization in training neural networks.

Even in the simple matrix factorization problems, the comparisons with state-of-the-art results in this exact setting are not clear to me.

**Questions:**

Line-5 on Page 1 and the other places, ''problem 1'' should be "Problem (1)".

---

> ### Author Response · Authors · 2023-11-16
> **Rebuttal for Reviewer yiZg**
>
> We thank the reviewer for the valuable suggestions and questions. In this rebuttal, we will address them point by point.
>
> - The presentation could be further improved
>     - In our updated version, we have significantly improved our presentation by taking all reviewers’ suggestions into account. Specifically, we summarize our results into a main theorem and an intuitive one. We move visualization and intuitive ideas to the introduction. Also, we delay detailed analysis to a section called sketch of proof. Finally, we add more comparisons to existing literature to emphasize our contributions.
> - The authors claim that they aim to understand the convergence of GD with large random initialization on simple matrix factorization problems, but as a reader, I could not find a main theorem or corollary that clearly states that with large random initialization, GD converges with certain rates under appropriate parameter settings and assumptions.
>     - In our revised version, we present an improved version of the main theorem, Theorem 6, in Section 4.2. We also provide an intuitive version of the theorem in the introduction. Our theorem is deterministic and applicable to large random initialization. In our theorem, we prove that all the quantities beyond $t^*_k$ are bounded by either a constant or a logarithmic term. Hence, we claim that we provide a detailed trajectory analysis of GD. If we assume $t_k^*=O(\log(d))$, then GD achieves $\epsilon$-accuracy in $O(\log(d)+\log(1/\epsilon))$ iterations. This is probably the desired result the reviewer want to see. However, due to the challenges posed by large initialization, we are currently unable to verify such assumption. This is one of the limitation of the paper. In some sense, this also reflects the difficulty of the problem we are studying. On the other hand, even without this assumption, our results still reveal the incremental learning behavior of GD and many other characteristics of GD. We hope the reviewer could appreciate our attempts to tackle the challenging and meaningful problem.
> - The 'large initialization' is in fact the `large random initialization'. The authors consider GD with large random initialization, but the variance still depends on the dimension of the problem under consideration. Therefore, when the dimension d is relatively large, it reduces to the ``small" random initialization setting.
>     - To clarify for the reviewer, our perspective on large initialization refers to the scenario where the norm of $X_0$ remains a positive constant, even as the dimension approaches infinity. By statistical concentration results, the norm of $N_0$ approaches to a finite positive constant when dimension d tends to infinity. Therefore, if $X_0=\varpi N_0$ with a positive constant $\varpi$ independent of d, then it belongs to the large initialization regime. On the other hand, if $\varpi$ tends to zero as d increases, then it belongs to the small initialization regime.
>     - In previous literature, $\varpi$ is as small as $\min(d^{-1/2},d^{-3\kappa^2})$, where $\kappa>1$ is the conditional number. In our case, $\varpi$ is a constant independent of d. As we discussed on page 6 below Theorem 6, we only require $\varpi=O(1/\sqrt{\eta})$ where $\eta$  is the step size. We also demonstrate that this is rate optimal. If we further increase it, the GD algorithm will simply diverge.
> - Even in the simple matrix factorization problems, the comparisons with state-of-the-art results in this exact setting are not clear to me.
>     - As we explained, there is a significant difference between large initialization and small initialization. Except the rank-one case, our work is the first one to study large initialization setting. This already makes the comparison clear. In addition, small initialization is in fact a special case of large initialization. All of our results hold for small initialization, too. Finally, large initialization is the one people are using in practice. Therefore, we hope the reviewer could appreciate our efforts to narrow the gap between theory and practice.
> - The authors consider a simple matrix factorization problem, but as claimed in the main text, the motivation of this paper is to better understand GD with large random initialization in training neural networks.
>     - Yes. Even for matrix factorization, the understanding of large initialization is rather limited. We hope our paper could bring people to notice this gap between theory and practice. We also hope our technique and results can inspire future research in related fields, including deep learning theory.
>
> If the reviewer has additional questions, please feel free to discuss with us.

---

> > ### Author Response · Authors · 2023-11-20
> > **Rebuttal discussion**
> >
> > Dear reviewer,
> >
> > We hope this message finds you well. We are writing to kindly follow up on the rebuttal we submitted for our paper. We have addressed the concerns and suggestions you raised during the review process and would greatly appreciate your feedback on the revisions we've made. Specifically, we have further clarified the contributions of our paper and the comparisons with previous results. Also, we have improved the presentation of the paper and you may find the revised paper in the platform. We are eager to hear your thoughts on the updated manuscript. Understanding your busy schedule, we would be grateful if you could take a moment to review our response at your earliest convenience. We would enjoy the opportunity to interact with you and answer any questions you might have after reacting to our response. Please let us know if you have any questions.

---

> ### Author Response · Authors · 2023-11-21
> **Rebuttal discussion**
>
> Hi Reviewer yiZg, we still hope to hear from you! In our understanding, you have two major concerns, and we have addressed them as follows.
>
> - First, the reviewer holds the point that we are considering small random initialization setting because the variance of each entry tends to zero as dimension $d$ increases.
>    - We would like to kindly point out that this argument is not correct. Suppose $N_0\in\mathbb{R}^{d\times r}$ with i.i.d. $N(0,1/d)$ entries. Then by standard concentration results, the spectral norm of $N_0$ will converge to a finite positive constant as dimension $d$ increases. If we do not require the variance of each entry tends to zero, such as taking $N_0$ as a matrix of $N(0,1)$ entries, then the norm of $N_0$ will explode and the GD algorithm will diverge when initialized with $N_0$. For the above reason, it is suitable to fix $N_0$ and define large or small initialization based only on the initial scale $\varpi$.
>    - There is a significant difference between our work and previous works. Prior works require $\varpi\lesssim\min(d^{-1/2},d^{-3\kappa^2})$ while we only need $\varpi\lesssim 1/\sqrt{\eta}$, which is dimension-independent. In addition, our condition is rate-optimal, meaning that a further increase of $\varpi$ would lead to divergence of the GD algorithm. Furthermore, one may review small initialization settings as special cases of our settings. As a result, the setting in our paper is much more challenging than previous results, and we provide a nontrivial extension of previous results.
> - Second, the reviewer suggests us to present a more concise and clear theorem.
>    - Thanks for the suggestion! We have carefully addressed this point. We have presented a clearer theorem that emphasizes our assumptions and results. Specifically, when using random initialization, our trajectory analysis holds almost surely. Moreover, if an additional assumption is made, we may obtain the $O(\log(d)+\log(1/\epsilon))$ convergence rate of GD. We have also included detailed discussions in the paper. Hope this revision can address the reviewer's concern.
>
> Please let us know if you have additional concerns. We are looking forward to your reply!

---

### Official Review · Reviewer_dfta · 2023-11-10

**Soundness:** 3 good
**Presentation:** 3 good
**Contribution:** 2 fair
**Rating:** 6
**Confidence:** 2

**Summary:**

The paper presents a stage-by-stage analysis of the dynamics of gradient descent on a low-rank symmetric matrix factorization problem, showing that the empirically observed stages of fast decrease followed by crawling progress before another fast decrease can be captured theoretically.

**Strengths:**

Fig. 1 and the first paragraph of 4.2 form a really nice motivation! I'd recommend moving them to the introduction. This forms a well-defined problem that is of relevance to understand the dynamics of gradient descent on more complex systems, which is relevant to the community. The approach appears novel, although I am not too familiar with the related work.

Altough the theory might be described as "incomplete", as it relies on an Assumption 8 to transition from learning the first component to the second, I appreciate that the submission is clear in that it is an assumption and gives a justification.

**Weaknesses:**

Up to minor clarity points below, the paper is understandable, but dense. My main concern regarding the submission its intended audience might be limited to the people looking to build upon those results to get towards a better understanding of symmetric matrix factorization. But my perspective is likely limited and there might be a wider applicability to the presented results.

**Questions:**

**Questions**
- The introduction states that the focus of the submission is on the implicit bias of gradient descent, but I do not see how studying the dynamics of Fig. 1 connects to the implicit bias (which, in my understanding, refers to the limit point the optimizer converges to)?
- Remark 2 states the the results hold for PSD $\Sigma$, but the results seem to also assume that $\Sigma$ is diagonal. Is this assumption necessary, or is it presented for the diagonal case wlog? What is the key difficulty in generalizing it to non-diagonal matrices?

**Minor points**
- Please define incremental to avoid confusion with the alternative use of incremental learning as a synonym for seeing-one-example-at-a-time learning. I realise in post that this is what the second-to-last paragraph in §1 is doing, but I didn't read it as such. A more explicit phrasing the first time the term appears might help, eg in the 4th paragraph "Jin et al. demonstrate an increamental learning phenomenon with small initialization; Eigenvectors associated with large eigenvalues are learned first". A sentence as to how this differs qualitatively from a similar observation on linear regression might help contextualize too.
- The introduction describes matrix factorization as "mirroring the training of a two-layer linear network", but this doesn't hold for the symmetric matrix factorization studied here, which seems more similar to quadratic regression
- The term "period" on page 7 might be replaced by "time", as the behavior is not periodic
- "ascend to infinity" is more commonly referred to as "diverge" or "diverge to infinity".

---

> ### Author Response · Authors · 2023-11-16
> **Rebuttal for Reviewer dfta**
>
> We thank the reviewer for the valuable suggestions and questions. In this rebuttal, we will address them point by point.
>
> - Fig. 1 and the first paragraph of 4.2 form a really nice motivation! I'd recommend moving them to the introduction.
>     - Thanks for the suggestion. We have moved them to the introduction in our updated version.
> - My main concern regarding the submission its intended audience might be limited to the people looking to build upon those results to get towards a better understanding of symmetric matrix factorization. But my perspective is likely limited and there might be a wider applicability to the presented results
>     - Symmetric matrix factorization intersects with a diverse array of problems such as compressed sensing, phase retrieval, matrix completion, PCA, quadratic regression, and neural networks, among others. It serves as a fundamental model that facilitates theoretical analysis. In addition, our work is the first one (except for the rank one case) to study large initialization. Such initialization is commonly used in practice, including neural networks or simply phase retrieval and PCA. Therefore, we expect our theoretical investigation will spark further research across these related fields.
> - The introduction states that the focus of the submission is on the implicit bias of gradient descent, but I do not see how studying the dynamics of Fig. 1 connects to the implicit bias (which, in my understanding, refers to the limit point the optimizer converges to)?
>     - We would like to offer two perspectives on the implicit bias of the gradient descent algorithm. Firstly, from the viewpoint of the limit point, the algorithm almost surely  converges to the global minimum. Although this is an optimization result, one may also view it as certain implicit bias, because the optimization problem is non-convex. Secondly, our trajectory analysis reveals incremental learning behaviors of GD during the training process. This second viewpoint, focusing on the training process, is typically useful when early stopping is utilized.
> - Remark 2 states the the results hold for PSD , but the results seem to also assume that  is diagonal. Is this assumption necessary, or is it presented for the diagonal case wlog? What is the key difficulty in generalizing it to non-diagonal matrices?
>     - There is no loss of generality to assume that $\Sigma$ is diagonal, because the analysis of GD is invariant to the orthogonal transformation. In our updated version, we have mentioned this point when we make the assumption. Moreover, our research can be easily extended to general symmetric (not necessarily positive semi-definite) cases.
> - Explanation of incremental learning.
>     - Thank you for the suggestions. In our revised version, we have explained that in our context, incremental learning refers to the process where eigenvectors associated with larger eigenvalues are learned first.
> - The introduction describes matrix factorization as "mirroring the training of a two-layer linear network", but this doesn't hold for the symmetric matrix factorization studied here, which seems more similar to quadratic regression
>     - The training of a two-layer linear network can be interpreted as asymmetric matrix factorization (MF). See Section 2.2 in Sun et al. (2019). While there are distinctions between asymmetric and symmetric MF, the study of symmetric MF can be seen as a vital preliminary step towards analyzing asymmetric MF. Moreover, we would like to mention that asymmetric MF with a suitable regularizer
>      $$\min_{X,Y}|\Sigma-XY^T|_F^2+\frac{1}{4}|X^T X-Y^T Y|_F^2$$
>     can be directly analyzed using the symmetric results.
> - Other minor points.
>     - We have updated our draft incorporating your valuable suggestions.

---

> > ### Comment · Reviewer_dfta · 2023-11-21
> > **Thanks for your response**
> >
> > Thanks for your answer.
> >
> > Some of the points raised in the responses would help readers if included in the main text, detailed below.
> >
> > I do not expect a response or a new version of the submission, those are only provided here as feedback.
> >
> > ---
> >
> > **Connection between the studied setting and NNets, and motivation for a broader audience**
> > As the disconnect between symmetric MF and neural networks has also been raised [by reviewer yiZg](https://openreview.net/forum?id=fAGEAEQvRr&noteId=M23bDOJ4nJ), an edit to this paragraph to make the motivation for the study of the asymmetric and symmetric MF (AMF, SMF) more explicit might help.
> >
> > The current introduction states that neural networks and AMF are similar, and the reader is supposed to fill in the gaps as to how the study of SMF connects to the behavior of neural networks. What was missing from my first read, and the reason I found Figure 1 compelling, is an argument along the lines of "incremental learning [short definition] has been reported in multiple models, including neural networks [citation]. This also happens in SMF, so let's try to describe this setting."
> >
> >
> > **Implicit bias**
> > > Firstly, [...] the algorithm almost surely converges to the global minimum[, which could be viewed as] implicit bias, because the optimization problem is non-convex.
> >
> > The limiting point does not qualify my working definition of implicit bias if the algorithm converges to _the_ (implicitly unique) global minimum, as the choice of optimizer does not impact the solution selected, as long as the algorithm converges. I would follow if the argument was referring to a specific choice of $X$ and the generalization property depended on $X$ rather than on $XX^T$, which is unique and doesn't depend on which solution is selected by the optimizer, but this doesn't seem to apply here.
> >
> > > Secondly, [...] incremental learning behaviors of GD during the training process [...] is typically useful when early stopping is utilized.
> >
> > The early stopping argument and its connection with the incremental learning idea is clearer. As the first point that comes to mind when the introduction talks about implicit bias, making this point explicit would help clear the confusion.
> >
> >
> > **Incremental learning**
> > The edit makes it clearer. However, the current description that "Eigenvectors associated with larger eigenvalues are learned first" seems incomplete. For example it also applies to a linear regression/convex quadratic functions (the error associated with the $i$th eigenvector evolves as $(1-\alpha\lambda_i)^t e^0_i$ where $\alpha$: step-size, $\lambda_i$: $i$th eigenvalue, $e^0_i$: initial error associated with the $i$th eigenvector). The penomenon studied here is, presumably, qualitatively different? For example, Gissin et. al (2020) describe it as "the singular values of the model are learned separately, one at a time", a component that seems missing from the current description.

---

### Author Response · Authors · 2023-11-16
**Rebuttal**

Thank you to all the reviewers for their valuable suggestions and questions. According to your suggestions, we have made the following modifications in our updated version.

- First, we have summarized our main results in a main theorem as well as an intuitive theorem. We put the intuitive theorem in the introduction. In particular, we emphasize two points. First, we establish a detailed trajectory analysis, and demonstrate the incremental learning phenomenon of GD. Second, we show that under an unverified assumption, the GD sequence achieves $\epsilon$-accuracy in $O(\log(d)+\log(1/\epsilon))$ iterations. Discussions are given for the unverified assumption.

- Second, according to reviewer dfta, we have moved the original experimental example and the related discussions to the introduction. This could help readers understand the intuitions of our results and analysis.

- Third, we move the rank-two matrix approximation part to a new section called Sketch of Proof. The main theorem is presented before this section. Therefore, readers will first see the main theorem and then see the detailed analysis.

- Fourth, we also address the reviewer MbHg’s concern about the structure of the original Section 3, challenge of general rank problems. In fact, in our updated version, we separate this section into two parts. One is the benign initialization, where we discuss the usage of SNR analysis for the local linear convergence. The other is the random initialization, where we review small initialization and also present the main theorem for large initialization. We hope this new structure is better.

- Besides the structural modifications, we also add contents to address reviewers’ concerns.
    - First, we additionally compare small and large initialization. Specifically, large initialization means the norm of $X_0=\varpi N_0$ tends to a positive constant when d tends to infinity. Small initialization means such norm converges to zero. By statistical concentration results, we know the norm $|X_0|=O(\varpi).$ Therefore, large and small initialization depends on whether $\varpi$  tends to zero as d increases to infinity. Moreover, to emphasize the difference, we review the small initialization condition in previous works. In particular, they require

         $$\varpi\lesssim \min\( \{d^{-1/2},d^{-3\kappa^2}\} \),$$

        where $\kappa>1$  is the conditional number. This condition converges to zero exponentially fast as d increases. In contrast, our paper requires $\varpi\lesssim 1/\sqrt{\eta}$, where $\eta$ is the step size. This is significantly looser than previous conditions. Moreover, we also demonstrate such condition is rate-optimal. This is because if the initial magnitude is larger, then the GD sequence may simply diverge. More detailed discussions and examples are provided in the updated version.

    - Second, we further explain the original assumption 8. In our updated version, we rewrite this assumption. We explicit introduce the variable $t^*_k$ and we consider two types of assumptions. The first assumption is that $t_k^*<\infty$. This is all we need for the trajectory analysis for GD, that is, the first six properties in Theorem 6 in the updated version. We demonstrate that this assumption almost surely holds when we employ random initialization. The second and stronger assumption is that $t_k^*=O(\log(d))$. We only need this assumption to show that GD achieves $\epsilon$-accuracy in $O(\log(d)+\log(1/\epsilon))$ iterations. Although we do not verify this assumption, we provide detailed discussions both below Theorem 6 and in Section 5.1.3.
    - Reviewer BGNj suggests us to conduct more experiments, we have provided additional experiments in various settings to further illustrate the results. However, due to the page limitations, we put these experiments in the final section of the appendix. In addition, we want to emphasize that our paper is a theoretical paper and our original experiment aims to provide intuitions. The original experiment in fact serves this purpose well.
    - We have also made other revisions to address reviewers’ other concerns. This includes the explanations of incremental learning, spectral method, assumption on $t_k^*$, the diagonal assumption of $\Sigma$, techniques of the SNR analysis. We have also fixed all the typos and added necessary references.
    - Finally, for some misunderstandings of our paper, we will discuss them point by point in our response to the corresponding reviewers.

Finally, if reviewers still have other concerns, feel free to discuss with us.

---

### Meta-Review · Area_Chair_YdLH · 2023-12-04

**Metareview:**

Reviewers were generally lukewarm about the broad applicability of the paper on symmetric matrix factorization and its interest to the ICLR community. I encourage the authors to consider a more general matrix factorization framework for greater generality and to compare to state-of-the-art techniques that employ other types of initialization methods.

**Justification For Why Not Higher Score:**

The applicability of the current framework is rather limited.

**Justification For Why Not Lower Score:**

Even though the scope of the paper is limited, some reviewers found the results interesting.

---

### Decision · Program_Chairs · 2024-01-16

Reject